# TRANSFORMER IS INHERENTLY A CAUSAL LEARNER

## ABSTRACT

We reveal that transformers trained in an autoregressive manner naturally encode time-delayed causal structures in their learned representations. When predicting future values in multivariate time series, the gradient sensitivities of transformer outputs with respect to past inputs directly recover the underlying causal graph, without any explicit causal objectives or structural constraints. We prove this connection theoretically under standard identifiability conditions and develop a practical extraction method using aggregated gradient attributions. On challenging cases such as nonlinear dynamics, long-term dependencies, and non-stationary systems, this approach greatly surpasses the performance of state-of-the-art discovery algorithms, especially as data heterogeneity increases, exhibiting scaling potential where causal accuracy improves with data volume and heterogeneity, a property traditional methods lack. This unifying view lays the groundwork for a future paradigm where causal discovery operates through the lens of foundation models, and foundation models gain interpretability and enhancement through the lens of causality.

## 1 INTRODUCTION

The ability to discover causality is foundational to intelligence, enabling understanding, prediction, and decision-making about the world. As an evolving field, causal discovery aims to formalize theoretical frameworks for identification criteria and to propose search algorithms to find the true causal structure from observational data (Pearl, 2009; Spirtes et al., 2000). In this area, causal discovery from time series focuses on identifying temporal causal dynamics by exploiting the temporal ordering that naturally constrains the direction of causation. Granger causality (Granger, 1969; Tank et al., 2021; Nauta et al., 2019) formalizes this intuition: a variable $X$ Granger-causes $Y$ if past values of $X$ contain information that helps predict $Y$ beyond what is available from past values of $Y$ alone. Additional methods extend this foundation, including constraint-based approaches like PCMCI and its variants that iteratively test conditional independence to examine the existence of causal edges (Runge et al., 2017), score-based methods like DYNOTEARS (Pamfil et al., 2020) that optimize graph likelihood with structural prior regularizations, and functional approaches like TiMINo and VAR-LiNGAM that leverage structural equation models and non-Gaussianity for identifiability (Peters et al., 2014; Hyvärinen et al., 2010).

Real-world systems exhibit complex interactions among many variables. For example, financial markets are highly non-stationary and involve very large variable sets (Engle, 1982); neural recordings exhibit strongly nonlinear population dynamics (Breakspear, 2017); climate sensor networks display long and short-term teleconnections (Wallace & Gutzler, 1981; Newman et al., 2016); and unstructured modalities such as video require modeling long-range spatiotemporal dependencies (Bertasius et al., 2021; Arnab et al., 2021). Despite rigorous theoretical foundations, prevailing algorithms are often constrained in practice by complex heuristics. Specifically, constraint-based and score-based approaches scale poorly: the number of statistical tests grows rapidly with dimension and lag, and non-parametric tests are computationally expensive (Runge et al., 2017; Chickering, 2002). Optimization approaches require careful tuning to achieve the right balance between likelihood and structural regularization (Zheng et al., 2018; Ng et al., 2020a; Pamfil et al., 2020; Zheng et al., 2019). More fundamentally, these estimators are not scalable representation learners: their learning is not transferable and thus offers little generalizability for zero-shot or few-shot adaptation; their effective capacity and expressiveness are not well-suited for pretraining on diverse systems.

Motivated by the striking performance and scaling behavior of autoregressive foundation models (Brown et al., 2020; Kaplan et al., 2020; Hoffmann et al., 2022), we ask whether the properties that make transformers strong forecasters can help causal discovery. Building this connection is valuable in two directions: for discovery, it promises data efficiency by leveraging pretrained representations and a scalable learning paradigm suited to complex dependencies; for foundation models, causal principles offer ways to diagnose limitations in memory and hallucinations, and guide architecture and objective choices. In this paper, we take a first step toward these goals: we revisit common identifiability assumptions in lagged data generation processes and show how decoder-only transformers trained for forecasting, together with input–output gradient attributions via Layer-wise Relevance Propagation (LRP) (Achtibat et al., 2024; Bach et al., 2015), reveal lagged causal structure. This view turns modern sequence models into practical, scalable estimators for temporal graphs while opening a path to analyze and strengthen foundation models through causal perspectives.

## 2 BACKGROUND

In this section, we start from reviewing causal discovery methods for time-series data, and the interpretability work of transformers, mainly focusing on language modeling tasks. Then we introduce our main motivations for connecting these two fields.

### 2.1 RELATED WORK

**Time-series Causal Discovery.** Recovering causal dynamics from temporal observations requires structural assumptions that determine when and how the true causal graph is identifiable. Constraint-based methods, assuming causal sufficiency and faithfulness, prune spurious edges via conditional independence tests (Entner & Hoyer, 2010; Runge et al., 2017; Malinsky & Spirtes, 2018). PCMCI and its variants extend the PC algorithm to handle nonlinear, contemporaneous, non-stationary, and high-dimensional settings (Runge, 2020; Martínez-Sánchez et al., 2024; Saggioro et al., 2020). Score-based methods search for structures that optimize scores encoding functional form and complexity preferences (Friedman et al., 2013; Nodelman et al., 2012), with recent continuous relaxations reducing computational cost (Sun et al., 2021; Pamfil et al., 2020). Functional methods exploit noise asymmetry to resolve edge orientation: VAR-LiNGAM assumes linear dynamics with non-Gaussian noise (Hyvärinen et al., 2010), while TiMINo generalizes to nonlinear additive-noise processes (Peters et al., 2013). Granger causality tests whether past values of one series improve forecasts of another, reflecting predictive rather than structural causation (Granger, 1969); neural extensions handle nonlinear dependencies (Tank et al., 2021; Nauta et al., 2019; Lu et al., 2023).

**Transformer Interpretability.** The success of large-scale pretrained transformers has motivated diverse interpretability efforts (Hewitt & Manning, 2019; Cunningham et al., 2023; Panickssery et al., 2023; Clark et al., 2019; Wang et al., 2022; Sundararajan et al., 2017). Most relevant to our work are methods for input–output token attribution in autoregressive models. Early work equated attention with explanation (Xu et al., 2015; Choi et al., 2016; Yang et al., 2016; Feng et al., 2018), but attention proves manipulable and misaligned with perturbation effects (Jain & Wallace, 2019; Serrano & Smith, 2019a; Bastings & Filippova, 2020). Post-processing techniques like attention rollout aggregate across layers and heads, yet remain forward-weight heuristics rather than faithful causal measures (Abnar & Zuidema, 2020). Gradient-based methods such as saliency, Integrated Gradients, and layer-wise relevance propagation estimate output-conditioned sensitivities through attention, residual, and MLP paths, yielding attributions that better align with perturbation tests (Sundararajan et al., 2017; Bach et al., 2015; Achtibat et al., 2024). Perturbation-based methods directly quantify importance via prediction changes under input erasure or counterfactuals (Li et al., 2016; Kokalj et al., 2021). We ground token-level attributions in identifiability theory for dynamic systems, providing a causally principled framework for interpreting transformers as structure learners.

### 2.2 MOTIVATIONS

Modern scientific and industrial systems are high-dimensional, nonlinear, non-stationary, and data-rich, precisely the regime where classical causal discovery faces brittle assumptions, exploding conditioning sets, and poor scaling with scenarios that have a large number of lags and variables,

and complex variable dependencies (Runge et al., 2019; Montagna et al., 2023). Decoder-only transformers are the opposite point in the design space: a single autoregressive objective, contextualized dependency modeling, and predictable gains from scale, data, and compute (Kaplan et al., 2020). Empirically, the same architecture transfers across modalities and domains and often outperforms specialized models, such as text, vision, and long-context time series (Liu et al., 2023; Liang et al., 2024). Our premise is pragmatic: if many real-world regularities arise from a small set of sparse, independent mechanisms, then an autoregressive learner that already scales and generalizes should be the natural interface to structure learning. This brings three immediate benefits: (i) amortization—forecasting supplies a ubiquitous training signal, turning structure learning into a unified and scalable manner, and we can build upon it according to the guidance of identifiability insights; (ii) the long-term potential of leveraging pretrained priors to reduce sample complexity when moving to new domains; and (iii) coverage—complex nonlinear, long-term dependencies and a diverse set of dynamic systems are handled within a unified framework.

On the other hand, framing decoder-only transformers through structure learning does two things. First, it suggests principled priors for analyzing and improving foundation models by aligning with the solid theoretical foundations of causal discovery, such as sparsity, modularity, and environment-invariance. Second, it clarifies what to measure: raw attention is not a faithful causal signal in deep stacks due to cross-layer mixing, whereas gradient/relevance-based criteria have stronger faithfulness guarantees and empirical support. The broader pay-off is a two-way bridge: causal principles guide foundation models toward stable, mechanism-level generalization, while foundation model practice delivers scalable, domain-agnostic causal discovery. This is a consolidation of where the empirical evidence already points and where causal theory has been heading.

## 3 A UNIFYING VIEW: IDENTIFICATION INSIDE ROBUST NEXT VARIABLES PREDICTION

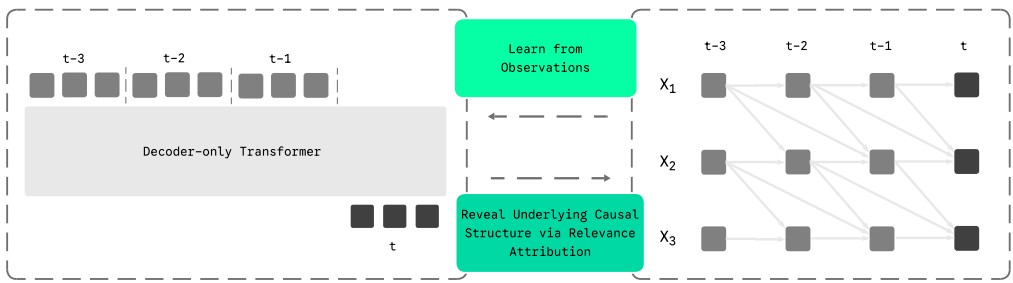

Figure 1: **Data generation and transformer-based causal discovery. Left:** A decoder-only transformer trained for next-step prediction. Tokens are lagged observations from $t-L$ to $t-1$; the model predicts $X_t$ from $X_{t-1:t-L}$. **Right:** A lagged data-generating process with $N = 3$ and window $L = 3$. Each $X_{i,t}$ depends on selected past values $X_{j,t-\ell}$ per the true graph $\mathcal{G}^*$. The trained transformer learns the process, and relevance attribution help recover the causal structure.

### 3.1 FROM PREDICTION TO CAUSATION

**Data-generating process.** Consider a $p$-variate time series $X_t = (X_{1,t}, \ldots, X_{p,t})^\top$ and a lag window $L \geq 1$. Each variable follows

$$X_{i,t} = f_i(\mathrm{Pa}(i,t), U_t, N_{i,t}),$$

where $\mathrm{Pa}(i,t) \subseteq \{X_{j,t-\ell} : j \in [p], \ell \in [L]\}$ are the lagged parents, $U_t$ are unobserved processes, and $N_{i,t}$ are mutually independent noises satisfying $N_{i,t} \perp (X_{<t}, U_{\leq t})$. We write $j \xrightarrow{\ell} i$ if $X_{j,t-\ell}$ is a direct cause of $X_{i,t}$. The lagged graph $\mathcal{G}^*$ contains $j \xrightarrow{\ell} i$ if and only if $X_{j,t-\ell} \in \mathrm{Pa}(i,t)$. The identifiability is defined as the unique recovery of the true causal graph $\mathcal{G}^*$ from observational data under the given assumptions. Note that this data generation process can model both linear and nonlinear relationships, as well as different types of exogenous noise and non-stationary dynamics.

This ensures the generality of our approach and the potential of applying this approach to a wide range of real-world scenarios.

---

**Assumptions for lagged identifiability**

- A1  Conditional Exogeneity.
- A2  No instantaneous effects (all parents occur at lags $\ell \geq 1$).
- A3  Lag-window coverage (the chosen $L$ includes all true parents).
- A4  Faithfulness (the distribution is faithful to $\mathcal{G}^*$)

---

Our identifiability result relies mainly on standard assumptions (A1–A4) commonly used in the causal discovery literature (Spirtes et al., 2000; Pamfil et al., 2020; Runge et al., 2017; White & Lu, 2010). We also assume the regularity of the data generation functions to avoid ill-posed conditions and guarantee the existence of the derivatives.

We briefly comment on the plausibility and practical remedies by combining traditional approaches when they are imperfectly met. *(A1) Conditional Exogeneity.* Causal sufficiency is a special case of A1 obtained by removing $U_t$; it permits latent confounders as long as they do not create spurious dependencies between targets and non-parents. When arbitrary latent variables are present, one can treat the learned structure as a Markov blanket and apply algorithms that handle latent variables (Malinsky & Spirtes, 2018) as post-processing. *(A2) No instantaneous effects.* If contemporaneous couplings exist, the analysis can be combined with algorithms capable of handling instantaneous effects (e.g., PCMCI+). We consider the variables that have instantaneous effects as latent confounders since they introduce spurious edges. Then we could easily handle this by using suitable algorithms like PCMCI+ with the initial skeleton consisting of the graph output from the transformer and a fully connected contemporaneous graph. The violation of *(A3)* can be abated by using a large window length, and *(A4)* holds generically. The unfaithful set (exact cancelations) in the linear Gaussian model has Lebesgue measure zero (Meek, 2013) and for nonlinear models, unfaithfulness becomes more restrictive. We recognize the lack of a mechanism for modeling latent variables and instantaneous effects as limitations of the current architecture and leave them as promising directions for future work.

---

**Causal Identifiability via Prediction**

**Theorem 1.** *Under A1–A4 and regularity conditions, the lagged causal graph $\mathcal{G}^*$ is uniquely identifiable via the score gradient energy: edge $j \xrightarrow{\ell} i$ exists iff $H_{j,i}^\ell := \mathbb{E}[(\partial_{x_{j,t-\ell}} \log p(X_{i,t} \mid X_{<t}))^2] > 0$.*

---

This theorem characterizes causal parents through sensitivity of the conditional distribution to each input. Unlike classical Granger causality, which tests mean prediction, the score gradient energy $H_{j,i}^\ell$ captures influence on the *entire* conditional distribution, including variance and higher moments. Under homoscedastic Gaussian noise, this reduces to $G_{j,i}^\ell := \mathbb{E}[(\partial_{x_{j,t-\ell}} f^*(X_{i,t}))^2]$, where $G_{j,i}^\ell > 0$ iff edge $j \xrightarrow{\ell} i$ exists. The estimator choice is flexible: any model that fits the conditional distribution suffices. We use decoder-only transformers for scalability and toward foundation model alignment. In practice, we approximate $G_{j,i}^\ell$ via Layer-wise Relevance $\tilde{G}_{j,i}^{(\ell)} := \mathbb{E}[\,|R_{ij}^{(\ell)}(X)|\,]$, then calibrate to recover $\mathcal{G}^*$. See Appendix §A.2 for the proof and §A.4 for the LRP–gradient connection.

## 3.2  Transformers inherit causal identifiability

We connect Theorem 1 to decoder-only transformers and make explicit why this architecture aligns with the identifiability program in Section 3.1, and how we extract a graph in practice. The connection has four parts: (i) alignment with assumptions A1–A4 and the forecasting objective, (ii) scalable sparsity and conditional-dependence selection, (iii) contextualized parameters for heterogeneity, and (iv) a structure extraction and binarization procedure.

**Alignment with identifiability and objective.**  We use a decoder-only transformer on a length-$L$ window. For each $t > L$, the input $\mathbf{s}_t = [X_{t-L}, \ldots, X_{t-1}] \in \mathbb{R}^{L \times p}$ is flattened to $L \cdot p$ tokens. We

use separate learnable node embedding and time embedding to distinguish the temporal dimension and node entities. Causal masking and autoregressive decoding enforce temporal precedence (A2); the window $L$ bounds the maximum lag (A3). We assume there are no hidden confounders (satisfying A1). Note that unlike traditional structure learning approaches that use a fixed input length to predict the last token, aligns with the standard autoregressive training that every token can be used as a training signal in teacher forcing. This benefits from the capacity and expressiveness introduced by the attention mechanism, where it naturally learns to fit all distributions from $p\left(X_t \mid X_{t-1}\right)$ to $p\left(X_t \mid X_{t-1:t-L}\right)$ at the same time. We optimize:

$$\min_{\theta} \; -\frac{1}{(T-L)\,L} \sum_{i=1}^{T-L} \sum_{k=1}^{L} \log p_\theta(X_{i+k} \mid X_{i:i+k-1}) \; + \; \lambda\,\Omega(\theta). \tag{1}$$

where $p_\theta(\cdot \mid \cdot)$ denotes the conditional likelihood parameterized by transformer outputs $\widehat{f}_\theta$ : $\mathbb{R}^{L \times p} \to \mathbb{R}^p$. For simplicity, we use a Gaussian likelihood (MSE objective), and $\Omega(\theta)$ is optional (e.g., sparsity regularization or entropy regularization; by default we do not use structural penalties).

**Sparsity and scalable dependence selection.** While explicit sparsity is not required for identifiability in the population, finite-sample recovery benefits from sparsity for both accuracy and efficiency. Constraint-based and score-based approaches control complexity via combinatorial conditioning and structural penalties, which limits scalability in high dimensions and long lags. Transformers implicitly sparsify: finite capacity, weight decay compress high-dimensional observations into generalizable parameters; softmax attention induces competitive selection among candidates (Martins & Astudillo, 2016; Sutton et al., 1998); and multi-head context supports selecting complementary parents. These priors make transformers well suited for scalable causal learning and can be complemented with explicit sparsity if desired.

**Attention as contextual parameters.** Attention matrices are input-conditioned and therefore act as contextualized parameters of pairwise dependencies rather than fixed population-level graph weights commonly used in optimization-based estimators (Zheng et al., 2018; Pamfil et al., 2020). Unlike methods that learn a single static binary mask, input-conditioned attention adapts to heterogeneity and non-stationarity: different contexts (time, regime) induce distinct effective dependency patterns. This is powerful and critical for learning multiple distinct dynamics at the same time, allowing the causal structures to have different sparsity, functional forms, and maximum lags. It is desirable and scalable in practice, enabling a data-driven mixture-of-graphs view without committing to a single mask.

**Structure extraction.** After training, we recover the structure via population gradient energy rather than raw attention. We use LRP (Achtibat et al., 2024) to compute relevance scores $R_{ij}^{(\ell)}$ that quantify the influence of variable $j$ at lag $\ell$ on predicting variable $i$ at time $t$:

$$R_{ij}^{(\ell)} \;=\; \sum_{m=1}^{M} \sum_{h=1}^{H} \mathrm{LRP}^{(m,h)}\big(\widehat{f}_\theta, X_t^{(i)}, X_{t-\ell}^{(j)}\big). \tag{2}$$

We aggregate these attributions across samples to estimate gradient energy $\tilde{G}_{j,i}^{(\ell)} = \mathbb{E}[\,|R_{ij}^{(\ell)}(X)|\,]$ and then calibrate to a sparse graph. Note that we do not use raw attention weights as causal explanations, since deep token mixing often misaligns attention scores with input and output dependence (Jain & Wallace, 2019). See Appendix §A.4 for implementation and aggregation details.

**Graph binarization.** We propose two rules to binarize it: (i) *Top-k per target:* for each target variable (row), select the $k$ largest entries as parents; this directly controls graph density and stabilizes precision. (ii) *Uniform-threshold rule:* assume a uniform baseline over $L \times p$ candidates and select entries whose normalized relevance exceeds $\frac{1}{L \times p}$. The two rules behave similarly at small scale; as context length grows, the uniform-threshold rule tends to degrade in F1 score compared to Top-$k$.

**Why gradients rather than raw attention.** Tokens in deep transformer layers are highly contextualized and are heavily mixed by downstream value projections and residual paths. Consequently,

large attention on a token does not guarantee a large effect on the target, and deep mixing can obscure true dependencies (Jain & Wallace, 2019). In contrast, gradients directly quantify local sensitivity of the target to each input coordinate; integrating their magnitude over the data yields a population-level measure aligned with our identifiability results. To obtain stable gradient attributions, we adopt Layer-wise Relevance Propagation (LRP), which conserves relevance through nonlinear blocks and reduces gradient noise while accounting for both attention and MLP pathways (Achtibat et al., 2024).

# 4 EXPERIMENTS

## 4.1 SIMULATION EXPERIMENTS

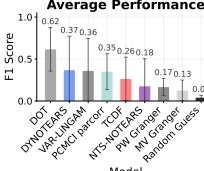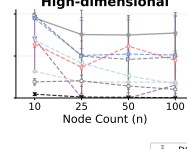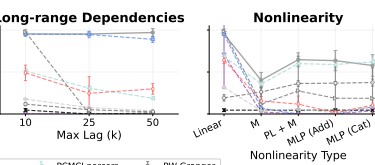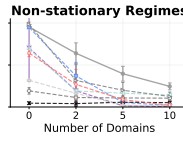

Figure 2: **F1 score analysis across regimes. (A)** Mean F1 across all experiments (averages exclude timeout cases). **(B)** High-dimensional input: F1 averaged across scales and seeds vs. the number of nodes. **(C)** Long-range dependencies: F1 averaged across scales and seeds vs. maximum lag. **(D)** Nonlinearity: F1 averaged across scales and seeds vs. different types of functional forms. **(E)** Nonstationarity: F1 averaged across scales and seeds vs. the number of domains. We run each method with three seeds. Missing results indicate method timeouts due to computational limits. DOT stands for Decoder-only Transformer. PL and M stand for piecewise linear and monotonic functions.

**Setup.** We evaluate our procedure for causal discovery using the simulator detailed in Appendix §A.6. We construct datasets from multiple dimensions including high-dimension, long-range dependency, nonlinear variable interactions, nonstationary processes, unobserved latent variables, and different exogenous noise types. We compare against baselines from a diverse set of algorithm families including PCMCI (Runge et al., 2017), DYNOTEARS (Pamfil et al., 2020), VAR-LiNGAM (Hyvärinen et al., 2010; Peters et al., 2014), NTS-NOTEARS (Sun et al., 2021), TCDF (Nauta et al., 2019), pairwise/multivariate Granger tests (Granger, 1969) and a random guess baseline based on the ground truth density. Both TCDF and NTS-NOTEARS are nonlinear methods. After training, we extract edges with LRP and binarize them with a per-target top-$k$ rule to get the inferred causal structure, and then examine the performance based on the F1 score. We also compare the variants of our approach including a shallow one-layer transformer, using attention as a dependency indicator, and different binarization methods (see Appendix A.6 for detailed experiment setups).

**General capabilities.** The transformer recovers lagged parents accurately and consistently across settings, achieving comparable or better performance to baseline methods (Figure 2A). The transformer maintains consistent and strong performance in all settings, where specialized approaches perform well in some settings but worse in others (Figure 2). Traditional methods degrade as dynamics and dimension grow, whereas the transformer remains robust without sensitive hyperparameter tuning. These advantages stem from the model's expressivity and attention-based dependency modeling. Performance improves steadily with sample size, making the approach suitable for complex real-world scenarios.

**Capabilities of modeling long-range and high-dimension dependencies.** The attention mechanism connects any pair of variables in one hop, making it excel at modeling long-range and complicated connections of a high-dimensional system (Vaswani et al., 2017; Likhosherstov et al., 2021). We examine this property for causal discovery with datasets including different maximum lags of causal effects and the number of input variables. Each setup has a linear and a non-linear version. Decoder-only transformers consistently surpass baselines in both high-dimensional and long-range dependency settings. Traditional algorithms like VAR-LiNGAM and PCMCI perform worse when

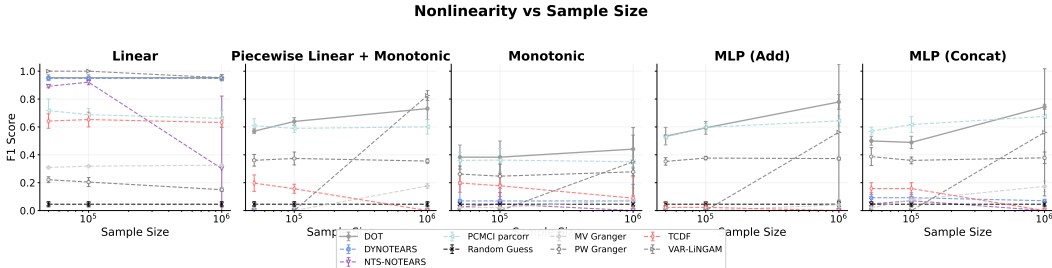

Figure 3: **Nonlinear dependencies.** F1 scores averaged across seeds vs. sample size in different nonlinear settings.

the input dimension increases, suffering from weaker detection power and the curse of dimensionality.

**Capabilities of modeling nonlinear interactions.** We examine the capability of the transformer in learning nonlinear interactions, considering settings from simple to complex: additive noise models with linear, monotonic, mixture of piecewise linear and monotonic, and multi-layer perceptron (MLP) as nonlinear functions, and non-additive noise models with MLP as mixing functions of variables and noise. We observe a trade-off between data efficiency and expressivity. While traditional methods employing simple estimators and search heuristics from human prior (e.g., DYNOTEARS, VAR-LiNGAM, PCMCI) can achieve good performance efficiently in simple cases like linear settings, a decoder-only transformer generally works better when the data scales and shows a consistent accuracy improvement as data increases.

Figure 4: **Non-stationary dependencies.** F1 scores averaged across seeds vs. sample size in different non-stationary settings.

**Scaling behavior in non-stationary settings.** The transformer can effectively leverage additional data to improve causal structure modeling accuracy. Here we construct two kinds of nonstationarity: the regular setting randomly samples linear structures with a fixed maximum lag for each domain, and the extreme setting randomly samples structure and nonlinear monotonic functions for each domain, within a range of maximum lags. Unlike traditional methods that become intractable with more data, the transformer shows consistent improvement across sample sizes (see Figure 4). In nonstationary settings, the model learns to handle multiple local mechanisms within a single framework. As the sample size increases, the transformer better separates and routes different causal structures corresponding to distinct regimes (Figure 2D). On the other hand, these learning curves along with nonlinearity experiment results show the limitation of weak structural priors and data hungriness. We see that when the number of domains increases, learning to accurately model and switch between them becomes much harder and requires much more data. This suggests that when data is insufficient relative to structural complexity, expressive models may resort to spurious correlations.

**Noise and latent variable robustness.** Transformers demonstrate robust performance across different noise distributions, maintaining consistent accuracy regardless of noise type or the variance properties of noise (see Figure 5). While we observe a performance drop of continuous optimization methods like DYNOTEARS and TCDF in non-equal noise variance settings aligned with Ng et al.

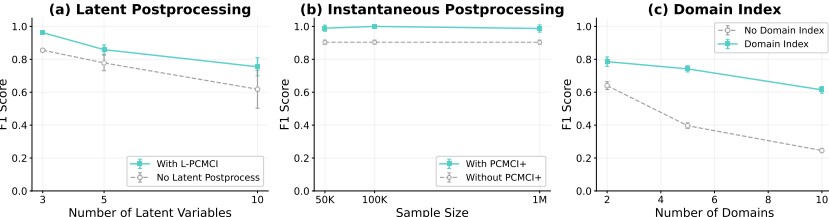

Figure 5: **Robustness to latent variables and noise. Left:** F1 scores on scenarios including different amounts of latent variables. **Right:** F1 scores on different kinds of noise (equal variance and non-equal variance)

(2024), the decoder-only transformer remains stable and accurate. However, due to the lack of a latent variable modeling mechanism, transformers are prone to learn spurious links and degrade as the number of latent variables increases, while traditional methods considering sparsity alleviate this influence.

Figure 6: **The potential of handling latent confounders, instantaneous relationships, and improving data efficiency by integrating known domain indicators.** When the assumptions of latent confounders and instantaneous relationships are violated, traditional causal discovery methods are adopted in postprocessing to effectively refine the learned structure. Integrating domain indicators improves data efficiency by helping recognize domain invariance and localize changes across domains.

**The potential of handling latent confounders.** Transformer performance degrades under latent confounding, and the architecture cannot generally model latent variables (see Figure 5). We show that it is possible to handle this by post-processing with a latent-aware causal discovery method: run L-PCMCI (Gerhardus & Runge, 2020) constrained by the transformer's predicted edges to refine the graph. Starting from the transformer's graph sharply reduces the expensive search space of latent-aware causal discovery methods. The combined pipeline is robust to latent confounders and yields substantially higher accuracy than the transformer alone (see Figure 6.a).

**The potential of handling instantaneous relationships.** The decoder-only transformer lacks the ability to model instantaneous relationships due to its autoregressive nature. One may consider unobserved contemporaneous input affects the autoregressive learning in decoder-only transformer like a latent confounder, which causes spurious edges in its internal causal structure. That being said, a similar approach to the one used for latent confounders can be employed to handle instantaneous relationships. We combine the learned structure from transformer with a fully connected contemporaneous graph and use it as the initial skeleton for a statistical causal discovery method that could handle both lagged and contemporaneous relationships, e.g., PCMCI+, DYNOTEARS (Pamfil et al., 2020; Runge, 2020). This procedure refines the confounded lagged graph and recognizes the instantaneous relationships, leading to a more precise recovered structure. Both contemporaneous input and unobserved confounders can be regarded as latent variables, which are common in the real-world data; however, the mechanism of handling them is lacking in current autoregressive learning.

We leave natural and native ways to model latent variables in transformers as a promising line of future work (see Figure 6.b).

**Integration with known domain indicators in non-stationary settings.** Exploiting variation across environments and distributions helps identify causal structures and representations (Huang et al., 2020; Khemakhem et al., 2020). Providing domain indicators lets the model separate cross-domain changes from invariants. We encode a domain index, proxying distribution shifts, as an additional input to a decoder-only transformer, improving data efficiency in both standard and highly complex settings and helping disentangle structure within representations (see Figure 6.c).

**Uncertainty analysis.** Statistical causal discovery outputs a population-level graph and estimates uncertainty via resampling (e.g., bootstrap). With transformers, we can aggregate per-sample point estimates and use their standard deviation to gauge consistency. Because larger mean relevance scores often have larger raw score variance, we rank each target's candidate parents within every sample and summarize these ranks by their mean and standard deviation. True edges show a higher mean rank and a lower rank standard deviation, indicating greater confidence (Figure 7). This offers a pragmatic way to surface the most reliable edges when precision is prioritized. In graphs with varied degrees, combining the mean and variance of ranks with a global top-k yields more accurate structures than using the mean of raw scores with row-wise top-k in both linear and nonlinear settings. More results are provided in Appendix §A.10.6.

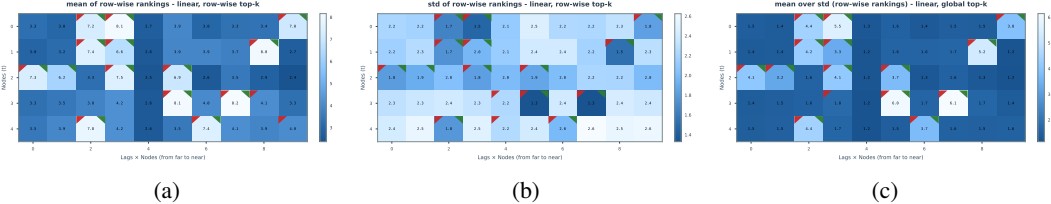

(a) (b) (c)

Figure 7: **Uncertainty analysis of causal structure estimation.** Mean and variance of relevance score rankings across samples for potential parents of target variables. Larger mean rankings tend to have a lower variance in rankings, indicating the model's confidence in identifying true causal relationships. The top-left red triangle means that model predicts there is a causal edge and top-right green triangle means that there is a true edge between the two variables.

**Attention and gradient attribution.** We also evaluate non-gradient proxies, such as raw attention scores, for recovering causal structure (see Figure 8). We see that the relevance between attention scores and gradient attributions differs for deep and shallow transformers. In deep transformers, attention scores barely reveal any information about the structure model learned, while in one-layer transformers, structures extracted from attention scores are much more accurate and aligned with the LRP's outputs. This aligns with findings that as depth increases, repeated attention routing and residual MLP updates mix token representations, making the attention not faithfully represent token dependency (Serrano & Smith, 2019b; Jain & Wallace, 2019).

**The effect of model depth.** The depth of the transformer primarily affects its capacity and expressivity to capture complex dependencies. With more layers, the transformer can model more complex structures and longer-range effects. In our nonlinear and long-range settings, deeper transformers achieve higher accuracy in recovering causal structure and show clear advantages with LRP readout. This highlights the potential of deep transformers for modeling highly heterogeneous long-range dynamics, echoing the success of pretrained large language and vision models.

**The effect of graph binarization.** Different binarization rules can lead to distinct causal graphs. We compare thresholding and top-$k$. Thresholding performs similarly to top-$k$ when the number of variables and the lag window are moderate, but its precision degrades as the context length grows. The importance of variables varies non-uniformly across lags with longer contexts. Top-$k$ provides a simple, effective way to control the precision–recall trade-off. Similar to max-depth limits in classical methods (PC, GES, etc.), choosing $k$ with domain knowledge lets us control edge density (e.g., use a small $k$ when the goal is to recover only the most important interactions).

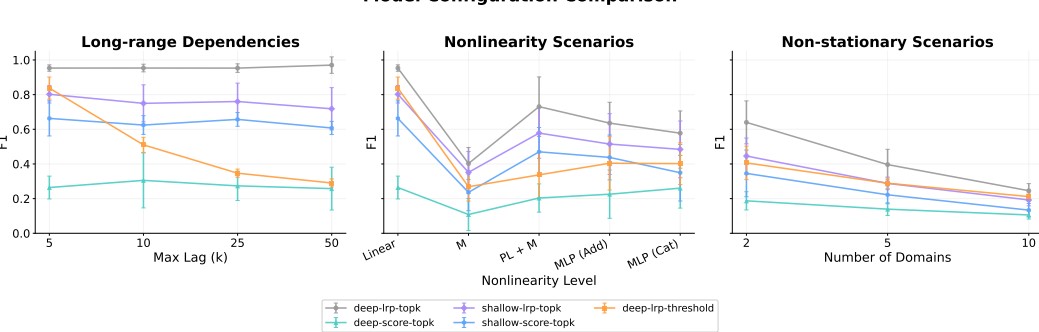

Figure 8: **Transformer variants performance comparison on challenging regimes. Left:** F1 scores on long-range dependencies. **Middle:** F1 scores on linear and nonlinear dynamics. **Right:** F1 scores on non-stationary dependencies.

## 5 CONCLUSION

We ask whether modern sequence models can reveal causal structure while learning forecasting. We prove that, under standard assumptions, decoder-only transformers trained autoregressively admit causal identifiability: the output's sensitivities to lagged inputs recover the true parents. We operationalize this with an aggregated gradient-energy readout (LRP) and a simple top-k binarization. Across nonlinear, long-range, high-dimensional, and non-stationary regimes, this procedure surpasses strong baselines and improves monotonically with data. This reframes discovery as a by-product of scalable representation learning while giving foundation models a causal lens. In sum, transformers are not only strong forecasters: read out with gradients, they are scalable causal learners, and causality, in turn, offers principled guidance to make foundation models more robust, data-efficient, and interpretable.

## 6 ETHICS STATEMENT

This work uses only synthetic time-series data generated by our simulator (Appendix A.6); no human subjects or personal/sensitive data are involved, and no IRB approvals were required. We adhere to the ICLR Code of Ethics. Our method recovers lagged causal graphs from observational data and relies on assumptions (conditional exogeneity, no instantaneous effects, adequate lag coverage, and faithfulness). We recommend expert review and, where feasible, interventional or quasi-experimental validation before deployment. Baseline implementations were used within their licenses and cited appropriately.

## 7 REPRODUCIBILITY STATEMENT

We provide a complete procedural description for reproducing results: identifiability proof in Appendix A.2, model and training objective in Equation 1 and Appendix A.7, attribution and graph extraction in Section 3.2, simulator and data-generation recipes (Appendix A.6), binarization rules (top-k and threshold), metrics (F1 with definition and evaluation protocol), and full hyperparameters, seeds, and splits (Appendix A.7). Baselines (PCMCI, DYNOTEARS, VAR-LiNGAM, TCDF, NTS-NOTEARS, pairwise/multivariate Granger) are documented with settings and versions. We shall release the code upon acceptance, and the submission includes sufficient detail for independent re-implementation of the pipeline and experimental setup.

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

# A  APPENDIX

## A.1  THE USE OF LARGE LANGUAGE MODELS (LLMS)

LLMs were used generically for polishing/clarity edits of prose, drafting code utilities (e.g., plotting and data-loading scaffolds), and initial figure/table layouts.

## A.2  IDENTIFIABILITY OF THE CAUSAL STRUCTURE

We formalize when gradients of the conditional distribution recover the lagged causal parents. Consider a $p$-variate time series where each variable follows

$$X_{i,t} = f_i(\mathrm{Pa}(i,t), U_t, N_{i,t}), \quad i = 1, \ldots, p,$$

where $\mathrm{Pa}(i,t) \subseteq \{X_{j,t-\ell} : j \in [p], \ell \in [L]\}$ are the lagged parents, $U_{i,t}$ are unobserved processes, and $N_{i,t}$ are independent noises. To identify the full causal graph, we analyze each target variable separately. Fix a target variable $Y := Y_t$ (which can be any $X_{i,t}$), and let $X = (X_1, \ldots, X_d)$ collect all covariates formed by stacking all $p$ variables over lags $1{:}L$. Write $S \subseteq \{1, \ldots, d\}$ for the index set of the direct time-lagged parents $\mathrm{Pa}(Y)$ inside $X$. For the target variable, we have:

$$Y_t = f_Y(\mathrm{Pa}(Y,t), U_t, N_{Y,t}), \qquad S = \text{indices of } \mathrm{Pa}(Y_t),$$

where $N_{Y,t}$ is independent noise satisfying $N_{Y,t} \perp (X, U_{\leq t})$.

**Assumptions.**  We work under the following conditions:

- **Conditional exogeneity**: the unobserved process satisfies $U_t \perp X_{S^c} \mid X_S$. This holds automatically when $U_t$ does not exist (becomes causal sufficiency) or is temporally independent (i.e., $U_t \perp U^{t-1}$), combined with no instantaneous effects (White & Lu, 2010).

- **No instantaneous effects**: edges from time $t$ to $t$ are absent; all parents of $Y_t$ live at lags $\ell \geq 1$ (Peters et al., 2013; Runge et al., 2017).

- **Lag-window coverage**: the constructed design vector $X$ contains all true lagged parents of $Y_t$ (the chosen maximum lag $L$ is at least the causal horizon).

- **Faithfulness**: the distribution is faithful to the underlying time-lagged graph, so that no independences arise from measure-zero cancelations (Pearl, 2009; Spirtes et al., 2000; Peters et al., 2013).

- **Support and regularity**: the law of $X$ admits a density supported on a rectangle $\Omega \subset \mathbb{R}^d$ (Evans, 2022; Adams & Fournier, 2003).

- **Conditional density and score regularity**: for $\mathbb{P}_X$-almost every $x \in \Omega$, the conditional law of $Y$ given $X = x$ admits a density $p^*(\cdot \mid x)$ with respect to Lebesgue measure on $\mathbb{R}$ that is strictly positive on its support. The log-density

$$\ell^*(y, x) := \log p^*(y \mid x)$$

  belongs to $W_{\mathrm{loc}}^{1,2}(\mathbb{R} \times \Omega)$ as a function of $(y, x)$, so that its weak partials $\partial_{x_j} \ell^*$ exist and are square-integrable under the joint law of $(Y, X)$.

Note that assuming conditional exogeneity is weaker than assuming causal sufficiency (no latent confounders) and independent noise (independent from past input and all the other exogenous inputs) at the same time: it permits latent confounders, provided they do not induce dependence between $Y_t$ and non-parents $X_{S^c}$ beyond what is mediated by the parents $X_S$. Moreover, the Causal Markov property $Y \perp X_{S^c} \mid X_S$ is not assumed separately; it follows from conditional exogeneity and the DGP. Here we assume plain faithfulness instead of strong-faithfulness (Uhler et al., 2013), which only requires the conditional independence in the probability distribution is entailed by d-separation in G. In other words, no 'accidental' independencies exist that aren't implied by the graph structure. The plain faithfulness violations are practically zero: (i). Unfaithful distributions form a collection of hypersurfaces in parameter space, which have Lebesgue measure zero (Meek, 2013; Spirtes et al., 2000). This becomes harder when the data generation process is nonlinear,

where the asymmetry caused by the nonlinear functions makes cancellation even harder. In the real world, violations are structurally unstable since any perturbation to parameters restores faithfulness. Below, we provide a general proof suitable for almost arbitrary observation distributions.

**Score gradient energy.** Define the *score* of the conditional density with respect to $x$ as

$$s(y, x) := \nabla_x \ell^*(y, x), \qquad s_j(y, x) := \partial_{x_j} \ell^*(y, x),$$

and the corresponding *score gradient energy*

$$H_j := \mathbb{E}\Big[\big(s_j(Y, X)\big)^2\Big], \quad j = 1, \ldots, d.$$

Intuitively, $s_j(Y, X)$ measures how sensitive the entire conditional law of $Y$ is to perturbations of the coordinate $X_j$ at the realized pair $(Y, X)$, and $H_j$ aggregates this sensitivity over the population.

**Lemma 1** (Zero weak partial implies no dependence). *Let $f \in W^{1,1}_{\mathrm{loc}}(\Omega)$ on a rectangle $\Omega \subset \mathbb{R}^d$. If $\partial_{x_j} f = 0$ holds almost everywhere on $\Omega$, then there exists a measurable $h$ with $f(x) = h(x_{-j})$ almost everywhere. Conversely, if $f$ does not depend on $x_j$, then $\partial_{x_j} f = 0$ holds almost everywhere.*

*Proof.* Assume $\partial_{x_j} f = 0$ almost everywhere. Fix $x_{-j}$. For almost every line $t \mapsto (t, x_{-j})$, it yields $f(t_2, x_{-j}) - f(t_1, x_{-j}) = \int_{t_1}^{t_2} \partial_{x_j} f(s, x_{-j}) \, ds = 0$, so $f(t, x_{-j})$ is (a.e.) constant in $t$. Thus there is a measurable $h$ with $f(x) = h(x_{-j})$ a.e. Conversely, if $f$ does not depend on $x_j$, then its weak partial $\partial_{x_j} f$ is 0 almost everywhere. $\square$

**Lemma 2** (Conditional exogeneity implies Causal Markov). *Under the data generating process $Y = f_Y(X_S, U_t, N_{Y,t})$ with $N_{Y,t} \perp (X, U_t)$, and conditional exogeneity $U_t \perp X_{S^c} \mid X_S$, the conditional distribution of $Y$ given $X$ depends only on $X_S$:*

$$p^*(y \mid x) = p^*(y \mid x_S).$$

*In particular, $Y \perp X_{S^c} \mid X_S$.*

*Proof.* By the DGP, $Y = f_Y(X_S, U_t, N_{Y,t})$. For any measurable set $A$,

$$P(Y \in A \mid X = x) = \int P(f_Y(x_S, U_t, n) \in A \mid X = x) \, dP_{N_{Y,t}}(n),$$

where we used $N_{Y,t} \perp (X, U_t)$. By conditional exogeneity, $P(U_t \mid X = x) = P(U_t \mid X_S = x_S)$, so the integrand depends on $x$ only through $x_S$. Hence $P(Y \in A \mid X = x) = P(Y \in A \mid X_S = x_S)$. $\square$

**Lemma 3** (Zero score partial and conditional independence). *Under the conditional density and score regularity assumption, the following are equivalent for each $j \in \{1, \ldots, d\}$:*

1. *$\partial_{x_j} \ell^*(y, x) = 0$ for almost all $(y, x) \in \mathbb{R} \times \Omega$;*

2. *there exists a measurable $h$ such that $\ell^*(y, x) = h(y, x_{-j})$ almost everywhere;*

3. *the conditional density $p^*(y \mid x)$ does not depend on $x_j$, i.e., $p^*(y \mid x) = \tilde{p}(y \mid x_{-j})$ almost everywhere for some measurable $\tilde{p}$;*

4. *$Y \perp X_j \mid X_{-j}$.*

*In particular, $H_j = \mathbb{E}[(\partial_{x_j}\ell^*(Y,X))^2] = 0$ if and only if $Y \perp X_j \mid X_{-j}$.*

*Proof.* Note that $H_j = 0$ if and only if $\partial_{x_j}\ell^*(y,x) = 0$ almost everywhere. By Lemma 1, this holds if and only if $\ell^*(y,x) = h(y,x_{-j})$ for some measurable $h$, which is equivalent to $p^*(y \mid x) = \exp(h(y,x_{-j}))$ depending on $x$ only through $x_{-j}$. This factorization of the conditional density is the standard characterization of $Y \perp X_j \mid X_{-j}$. $\qquad\square$

**Theorem 1** (Score-based characterization of lagged parents). *Under the data generating process, conditional exogeneity, no instantaneous effects, Causal Markov and Faithfulness, and the conditional density and score regularity assumptions, the score gradient energy $H_j$ recovers the full lagged structural parent set:*

$$H_j = 0 \iff j \notin S.$$

*In particular, if $k \in S$ then $H_k > 0$.*

*Proof.* **Claim 1:** $j \notin S \Rightarrow H_j = 0$. By the Causal Markov property for the time-lagged graph, $Y_t \perp (\text{Past} \setminus \text{Pa}(Y_t)) \mid \text{Pa}(Y_t)$. In particular, $Y \perp X_{S^c} \mid X_S$. This implies $Y \perp X_j \mid X_{-j}$ for any $j \in S^c$. Lemma 3 then yields $H_j = 0$.

**Claim 2:** $H_j = 0 \Rightarrow j \notin S$. Suppose $H_j = 0$. By Lemma 3, this is equivalent to $Y \perp X_j \mid X_{-j}$. Consider the time-lagged causal graph in which $S$ indexes the direct lagged parents of $Y_t$ inside $X$. If $j \in S$, there is a directed edge $X_j \to Y$, and this edge is not blocked by conditioning on $X_{-j}$, so $X_j$ and $Y$ are d-connected given $X_{-j}$. Thus the graph does *not* entail $Y \perp X_j \mid X_{-j}$. By Faithfulness, such a conditional independence cannot hold in the observational distribution if $j \in S$. Hence $j \notin S$. For any $k \in S$, the contrapositive implies that $Y \not\perp X_k \mid X_{-k}$ and therefore $H_k > 0$. $\qquad\square$

## A.3 Special case: Homoscedastic additive Gaussian noise

We now specialize to the homoscedastic additive noise setting, where the score gradient energy reduces to a simple function of regression gradients. We use the regression-based gradient attributions and MSE objective in our implementations for simplicity.

**Additional assumption.**

- **Homoscedastic additive Gaussian noise**: the structural equation for $Y_t$ takes the form

$$Y_t = f(X_S) + \epsilon, \qquad \epsilon \sim \mathcal{N}(0, \sigma_0^2), \quad \epsilon \perp X,$$

where $f : \mathbb{R}^{|S|} \to \mathbb{R}$ is a measurable function with $f \in W^{1,2}_{\text{loc}}$ and $\sigma_0 > 0$ is a constant.

Under this model, the conditional distribution is $Y \mid X = x \sim \mathcal{N}(f(x_S), \sigma_0^2)$, with log-density

$$\ell^*(y,x) = -\frac{(y - f(x_S))^2}{2\sigma_0^2} + \text{const.}$$

Define the *gradient energy* of the regression function by

$$G_j := \mathbb{E}\left[(\partial_{x_j} f(X_S))^2\right], \quad j = 1, \ldots, d.$$

**Proposition 2** (Score gradient energy under homoscedastic Gaussian noise). *Under the homoscedastic additive Gaussian noise model, the score gradient energy satisfies*

$$H_j = \frac{G_j}{\sigma_0^2}.$$

*In particular, $H_j = 0$ if and only if $G_j = 0$.*

*Proof.* Differentiating $\ell^*(y, x)$ with respect to $x_j$ yields

$$\partial_{x_j}\ell^*(y, x) = \frac{y - f(x_S)}{\sigma_0^2} \cdot \partial_{x_j}f(x_S).$$

Define the standardized residual $Z := (Y - f(X_S))/\sigma_0$. By construction, $Z \sim \mathcal{N}(0, 1)$ and $Z \perp X$. Substituting $Y - f = \sigma_0 Z$ gives

$$\partial_{x_j}\ell^*(Y, X) = \frac{Z}{\sigma_0} \cdot \partial_{x_j}f(X_S).$$

Taking expectations and using $Z \perp X$ and $\mathbb{E}[Z^2] = 1$:

$$H_j = \mathbb{E}\left[\frac{Z^2}{\sigma_0^2}(\partial_{x_j}f)^2\right] = \mathbb{E}[Z^2] \cdot \mathbb{E}\left[\frac{(\partial_{x_j}f)^2}{\sigma_0^2}\right] = \frac{G_j}{\sigma_0^2}.$$

Since $\sigma_0 > 0$, we have $H_j = 0$ if and only if $G_j = 0$. When we assume the noise is sampled from the standard Gaussian distribution, $H_j$ turns to $G_j$. $\square$

**Corollary 3** (Gradient characterization under homoscedastic Gaussian noise). *Under the homoscedastic additive Gaussian noise model together with all assumptions from Theorem 1, the gradient energy $G_j$ recovers the full lagged structural parent set:*

$$G_j = 0 \iff j \notin S.$$

*In particular, if $k \in S$ then $G_k > 0$.*

*Proof.* By Proposition 2, $H_j = G_j/\sigma_0^2$ with $\sigma_0 > 0$. Thus $H_j = 0$ if and only if $G_j = 0$. Combining with Theorem 1 yields the result. $\square$

**Remark 1** (Extensions). *Analogous closed-form expressions for $H_j$ can be derived for other noise models (e.g., heteroscedastic Gaussian, non-Gaussian additive noise) by computing the score $\partial_{x_j}\ell^*$ and taking expectations. In heteroscedastic settings, $H_j$ generally decomposes into terms capturing the sensitivity of both the conditional mean and variance to $X_j$, ensuring that parents affecting only higher moments are correctly identified.*

**Remark 2** (Practical estimation of the score). *In practice, the choice of model for estimating $\ell^*(y, x)$ or $s_j(y, x)$ depends on prior knowledge about the conditional distribution. When Gaussian noise is assumed, minimizing mean squared error yields the conditional mean $f^*(x)$, and $G_j$ can be estimated from the squared gradients of the learned regressor. For categorical outcomes, cross-entropy loss corresponds to the negative log-likelihood, and the score is obtained by differentiating the logits with respect to $x$. When the conditional distribution is unknown or highly complex, one may employ flexible density estimators such as mixture density networks, normalizing flows (Rezende & Mohamed, 2015; Papamakarios et al., 2021), or continuous normalizing flows trained via flow matching (Lipman et al., 2022).*

For computational convenience and implementation simplicity, we consistently employ MSE loss as the training objective and use the regression gradient-based attributions as the score gradient energy, assuming exogenous input sampled from a standard Gaussian distribution.

A.4 ATTENTION LRP AS A SURROGATE FOR GRADIENT ENERGY

Layer-wise Relevance Propagation (LRP) decomposes a model's output $f(x)$ into relevance scores assigned to input coordinates. For efficiency and simplicity, we adopt the Input×Gradient formulation of $\varepsilon$-LRP, which expresses LRP as a single chain of Jacobian–vector products (one backward pass) with small, local modifications to the backward rule at nonlinearities and at attention/normalization layers. This implementation is equivalent to $\varepsilon$-LRP up to a layer-wise rescaling and closely follows the efficient Attention-LRP formulation used for transformers (Achtibat et al., 2024).

Concretely, for a trained forecaster $\hat{f}$ and a scalar prediction $z := \hat{f}(x)$ (e.g., the mean for regression or a logit/probability for classification), we define per-sample relevance by

$$R(x) := x \odot \widetilde{\nabla}_x z,$$

where $\widetilde{\nabla}_x$ denotes a gradient computed with the modified local Jacobians described below. Aggregating coordinates gives a global score

$$\tilde{G}_j \; := \; \mathbb{E}\big[\,|R_j(X)|\,\big],$$

used as a monotone proxy for $G_j = \mathbb{E}[(\partial_{x_j} f^*(X))^2]$.

**Core (Input×Gradient) LRP equations.** For computational efficiency, we use the gradient-input reformulation in attention-aware LRP (Achtibat et al., 2024):

$$R(x) \; = \; x \; \odot \; \Big( J_1 \, J_2 \, \cdots J_L \, e_i \Big) \qquad \text{(Input×Gradient with modified local Jacobians).} \quad \text{(IG-1)}$$

The same chain-of-Jacobian idea applies to attention and normalization layers in transformers. In practice, this yields LRP attributions in a single backward pass, after which token-level relevances are aggregated to $\tilde{G}_j$ as above.

## A.5 THE RELATIONSHIP BETWEEN INTERVENTION-BASED EFFECTS AND GRADIENT-BASED GRAPH ATTRIBUTIONS

Intervention is often adopted in real-world causal inference scenarios to measure causal effects, which works as a golden standard to recover causal relationships. We intervene in the input data by one standard deviation and measure the average. We analyze the relationship between them using both linear and nonlinear datasets (see Figure 9 and Figure 10), and we find that they are highly correlated. The more accurately the causal structure model learns (linear case), more aligned they are. This implies the consistency between the internal causal model and the causal graph learned in the decoder-only transformer. One could also use the perturbation tests to recover the causal graph learned in the transformer instead of using gradient attributions.

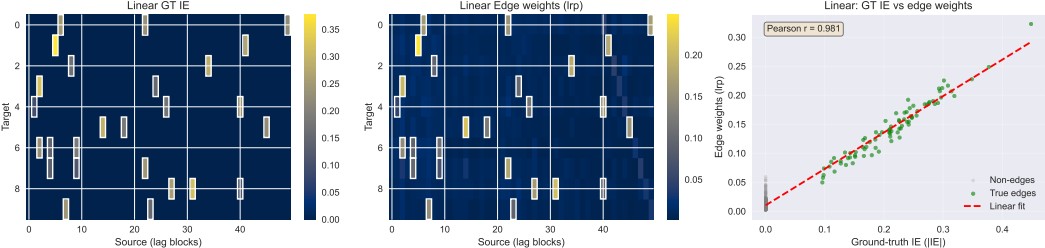

Figure 9: **Intervention effects and gradient-based attributions in linear case.** The intervention effect is strongly correlated with the relevance score in the linear scenario.

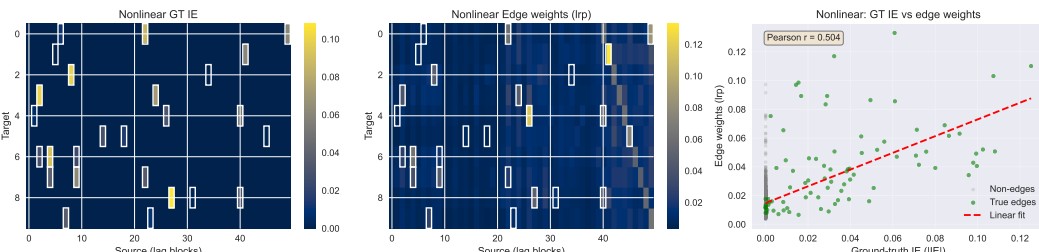

Figure 10: **Intervention effects and gradient-based attributions in nonlinear case.** The intervention effect is strongly correlated with the relevance score in the nonlinear scenario.

## A.6 EXPERIMENT SETUPS

### DATA GENERATION AND SIMULATION

**Simulator.** We use the CDML-NeurIPS2020 structural time-series simulator to sample datasets (Lawrence et al., 2020). We use a linear baseline and multiple variants in different dimensions such as number of variables, maximum lag, noise type, non-stationarity, latent variables. For the variants, we only vary the interested property of the data generation process compared to the linear baseline, and use multiple sample sizes to see how the performance changes with the sample size (5e4, 1e5, 1e6).

**Variables and lags.** For a system with $N$ observed variables and maximum lag $K$. We disable instantaneous effects and set the transition probability of 0.3. Latent and noise autoregression are set to 0 unless noted. We use the number of variables 10, 25, 50 to study the high-dimensional cases.

**Control graph density via expected in-degree.** To obtain comparable sparsity across $N$ and $K$, we specify an expected in-degree $E_{\text{in}} = 3$ per node (aggregated across all parent candidates).

**Structural functions and nonlinearity.** We control the nonlinearity complexity by employing functional forms as follows (first 3 are additive noise models): (1) piecewise linear (PL): mixture of linear, piecewise linear, (2) and monotonic (sum-of-sigmoids) functions (3) MLP (add): multi-layer perceptron (MLP) with additive noise injection (4) MLP (concat): MLP aggregation with noise concatenation.

**Noise types.** We consider three noise types: Gaussian (in linear baseline), Uniform, and Mixed. The mixed noise is a fixed mixture over distributions [Gaussian, Uniform, Laplace, Student's t]. We also study non-equal variance noise, with small range from 0.5 to 5 and large range from 0 to 10.

**Non-stationarity.** To study how different approaches behave under time-varying causal structure, we partition the sequence into $S$ contiguous segments ($S \in \{2, 5, 10\}$) and independently generate each segment with a randomly sampled graph. We construct two settings here, the first is the regular setting where each domain have the same maximum lag (5) and the data generation process are all linear. In the extreme setting, each domain might not have the same maximum lag (up to 5) and the data generation process are composed of monotonic functions.

**Latent variables.** We examine the robustness of discovery methods in the presence of latent variables. We set the number of latent variables to $L \in \{3, 5, 10\}$.

## A.7 TRAINING DETAILS AND MODEL ARCHITECTURE

We train autoregressive Transformers on lag-$K$ windows, after per-variable z-score normalization. We use an embedding dimension of 64, 4 attention heads, and either 1 ("shallow") or 4 ("deep") layers with pre-LayerNorm, residual connections, and a 2-layer ReLU feed-forward; causal masking, node/time embeddings. Models are optimized with Adam (learning rate 1e-3, batch size 256) under an MSE objective, with gradient clipping at 1.0. We use the official implementations for PCMCI (Runge, 2020), VAR-LiNGAM (Hyvärinen et al., 2010), TCDF (Nauta et al., 2019), NTS-NOTEARS (Sun et al., 2021), and Granger causality implementations from a collection repository of time-series causal discovery algorithms [Anonymous link]. The random guess baseline samples from a binary distribution according to the average incoming degrees, to acquire the lagged causal graph. We use the default hyperparameters from the official implementation.

## A.8 COMPUTE RESOURCES

All transformer experiments are implemented in PyTorch and executed in FP32 precision on a single NVIDIA A100 GPU, with actual memory usage below 24 GB. Experiments that exceed six hours of runtime, including both our transformer approach and baseline methods, are terminated and classified as timeouts.

## A.9 DISCUSSIONS

**Transformer-based causal discovery.** Recently, the transformer architecture has been proposed as a powerful alternative for discovering causal structures from data. CausalFormer combines additional modules such as multi-kernel causal convolution and a decomposition-based detector to extract the global causal graph from time-series data (Kong et al., 2024). Symmetry-aware transformers preserve temporal asymmetry for causal discovery in financial time series (Zheng & Liu, 2025). Besides unsupervised causal discovery, transformers are applied to learn a broad range of predefined causal relationships with synthetic data (Ke et al., 2022) and domain-specific causal graphs with interventional data (Wang & Kording, 2022) in a supervised manner. BCNP (Bayesian Causal Neural Process) meta-learns bayesian networks by training a transformer-like encoder-decoder architecture on synthetic data for priors and adjusting the posterior in test-time (Dhir et al., 2024). In this work, we establish the connection between prediction and structure identification through transformers, with theoretical guarantees that include required conditions and concrete representations to identify it. Without extra modifications, a decoder-only transformer naturally serves as a scalable causal learner in an unsupervised manner.

**Scalability in causal discovery.** Scalability is crucial for causal discovery algorithms to be applied in real-world scenarios. Factors such as sample size, variable size, graph density, functional forms, and data quality (e.g., latent confounders and missing values) require algorithm scalability. Previous search methods tackle the combinatorial search challenges in large graphs via computational optimization, such as device acceleration (Zarebavani et al., 2019; Akinwande & Kolter, 2024) and algorithm optimization techniques like divide-and-conquer and dynamic programming (Shah et al., 2024; Ramsey et al., 2017; Hyttinen et al., 2017). On the contrary, continuous-optimization and neural granger causality methods (Pamfil et al., 2020; Nauta et al., 2019; Tank et al., 2021) find causal graphs at model convergence within certain optimization steps, which are not strictly limited by the combinatorial search space. However, these methods are only trained on a homogeneous dataset and are able to recover the corresponding population-level graph. They do not allow for heterogeneity and are often jointly optimized with heuristics that are sensitive to the hyperparameters (e.g., L1). By connecting to the prediction-first framework through transformers, we should expect in the long run: 1) the appearance of a large-scale causal discovery model trained via other downstream tasks with some causal auxiliary loss, such as forecasting tasks; 2) the gradual utilization of foundation models for causal discovery, which may eventually capture causal relationships in a broad range of complex data distributions with few-shot or minimal finetuning. Moreover, the advancements in the tokenization of foundation models should improve the scalability further by compressing more high-dimensional input effectively.

**Structural causality and Granger causality.** Our framework is grounded in *structural causal models* (SCMs), where $X_j \rightarrow Y$ means $X_j$ appears in the structural equation for $Y$—a mechanistic notion independent of statistical tests (Pearl, 2009; Peters et al., 2017). White & Lu (2010) linked Granger and structural causality: under conditional exogeneity ($U_t \perp X_{S^c} \mid X_S$), Granger noncausality equals *structural noncausality almost surely*. However, without faithfulness, edge cases exist where true causes are undetectable, falling short of exact identification. We strengthen this in three ways. First, by imposing faithfulness, we obtain an *exact* characterization: $H_j = 0 \iff X_j \notin S$ (Theorem 1). Second, we provide a concrete criterion, the score gradient energy $H_j = \mathbb{E}[(\partial_{x_j} \log p(Y|X))^2]$, directly computable from learned densities. Third, we connect this with transformers, enabling scalable causal discovery. Under standard Gaussian noise, $H_j$ reduces to $G_j$ (Corollary 3), connecting to standard MSE training. Crucially, the logical direction is from structural causality to predictive dependence: when Granger-style criteria succeed, they do so as a consequence of the structural theory under identifying assumptions—not as an independent foundation.Thus, structural identification and distributional Granger causality converge under our assumptions, but the guarantees derive from the SCM, with Granger-style dependence as a consequence and not a foundation.

**Relationships to previous theory results.** Classical Granger causality (Granger, 1969) tests predictive improvement in linear VARs but lacks structural interpretation. Subsequent work pursued structural identification through diverse assumptions: LiNGAM (Hyvärinen et al., 2010) achieves full DAG identification via non-Gaussianity but requires linearity; PCMCI (Runge, 2020) performs conditional independence testing under causal markov, faithfulness and causal sufficiency;

DYNOTEARS (Pamfil et al., 2020) enables continuous optimization for linear time-series DAGs; Neural Granger Causality (Tank et al., 2021) extends to nonlinear functions but establishes only sufficient conditions. The closest antecedent is White & Lu (2010), who proved under conditional exogeneity ($D^t \perp U^t \mid Y^{t-1}, X^t$) that Granger non-causality is equivalent to structural non-causality almost surely—that is, structural causality may exist on measure-zero sets without manifesting in conditional distributions. Our Theorem 1 strengthens this by imposing faithfulness to eliminate such pathological cases, yielding an exact characterization: $H_j = 0 \Leftrightarrow j \notin S$. Moreover, while White & Lu (2010) relies on conditional independence testing, we provide a computable criterion, score gradient energy, that directly connects structural identification to the prediction objective, enabling practical extraction via gradient-based attribution in transformers.

**The difficulty of non-convex optimization.** Traditional continuous-optimization approaches to causal discovery struggle with non-convex loss landscapes. Even under identifiability and with the correct objective, nonconvexity resulting from unequal noise variances and nonlinearities can make structure recovery nearly intractable; outcomes hinge on fragile initialization, especially with limited homogeneous data (Ng et al., 2024). By contrast, large-scale transformer pretraining operates in a different regime: overparameterized networks have benign landscapes with many global minima (Du et al., 2019); in high dimensions, bad local minima are rare, while saddle points dominate (Kawaguchi, 2016); and the stochasticity of SGD helps escape saddles and favors flatter, more generalizable regions (Jin et al., 2017). This geometry enables transformers to function as scalable causal learners, effectively sidestepping the non-convex barriers that constrain classical methods.

**The role of prediction objective.** A Gaussian likelihood (equivalently, an MSE loss) corresponds to assuming additive homoskedastic Gaussian observation noise, but richer likelihoods can better capture heteroskedastic or multimodal dynamics and sharpen attribution. Objectives should match the data distribution and exogenous noise. For instance, adding degrees of freedom can accommodate unequal noise variances in highly heteroskedastic data. According to our identifiability results, promising alternatives include flow-matching and diffusion objectives, as well as quantile and energy-based losses; these better model stochasticity and complex distributions. As predictive fidelity improves, the implicitly learned structure should become more accurate.

**The need for better structure priors.** Our experiments indicate that vanilla decoder-only transformers are sample-hungry for recovering correct structures from a single nonlinear generator, and heterogeneous mixtures are harder—evidence of a weak inductive bias for causal structure. Causal theory offers mature priors to close this gap. Independent Causal Mechanisms and minimality enforce mechanism independence and modular factorization, yielding cross-environment invariances and improved sample efficiency (Peters et al., 2017; Huang et al., 2020). Sparsity further reduces the search space, making learning more tractable in noisy settings (Ng et al., 2020b; Zheng et al., 2018; Perry et al., 2022). Recent large-language model architectures echo these ideas: gated and block-sparse attention instantiate sparsity and modularity, mitigating spurious context coupling and improving long-context retrieval and robustness to distribution shift (Yuan et al., 2025; Lu et al., 2025). Finally, scalable, native modeling of latent variables and instantaneous effects broadens the class of structures beyond lagged processes. Integrating such priors should improve efficiency, generalizability, and robustness.

## A.10 Additional Experiments

### A.10.1 Can Current Time-Series Foundation Models Recover Structure in a Zero-Shot Way?

Here, we attempt to explore the possibility of directly using current foundation models in time-series forecasting for zero-shot causal discovery. We adopt Chronos2 Ansari et al. (2025), the latest state of the art time-series forecasting model in both univariate and multi-variate modeling. It is tested on three linear datasets for simplicity, with the same forecasting and graph extraction procedures as we use for the vanilla transformer. Due to the patch-based input and output nature of Chronos2, we are able to use a much longer input to include more information for prediction (predict the next patch (16 time steps) given previous 20 patches). We find that using a larger context length is more aligned with the training distribution and improves forecasting accuracy. In the Figure 11, we see that the zero-shot forecasting accuracy is good but not perfect, and corresponding structure inferred

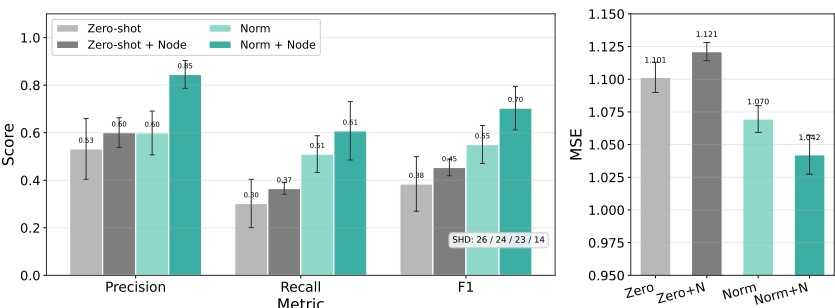

Figure 11: **Structure recovery with Chronos2 on linear data.** In zero-shot mode, Chronos2 uses a randomly initialized node-embedding layer for forecasting and structure extraction. Under parameter-efficient finetuning, only normalization layers and the new node-embedding layer are trained to adapt to unseen data and distinguish variables.

is suboptimal. We attribute this to mainly two reasons: firstly, the raw time-series is less structural, more noisy and more complicated compared to the other modalities like language, which makes it hard for the model to learn useful and generalizable inductive bias. Secondly, as we show in the nonstationary experiments, learning correct structures with mixed data is not very data-efficient. Even with large-scale pretraining, the effective sample size for the model to learn and understand dynamics is still not enough. We see that finetuning on the domain-specific data helps improve forecasting and causal discovery effectively. We also find that the identity modeling of variable is important but lacking in current multi-variate foundation models. Simply learning a node embedding layer helps the model to recognize variables, which benefits both forecasting and structure recovery. Even adding randomly initialized node embeddings without finetuning helps the model infer more accurate causal structure. We conclude that the current time-series foundation model can serve as a noisy zero-shot discoverer and might not be practical to directly adopt in real-world time-series applications. With advancements in time-series tokenization and learning methods for more universal inductive bias in the future, we should expect robust zero-shot forecasting and causal discovery capabilities from foundation models.

### A.10.2 PERFORMANCE UNDER NON-STATIONARY SETTINGS WITH REALISTIC MINIMAL CHANGES.

In real world settings, the changes across domains are often minimal and gradual. We construct a setting in which a small part of the structure is randomly rewired compared to the previous regime. We see that data efficiency is much higher compared to the randomly sampled setting in both regular and extreme settings (see paragraph 4.1). Note that here we do not inject any prior or constraint about minimal changes to the architecture, and we should expect it to be much more data-efficient when we incorporate such prior knowledge natively into the transformer.

### A.10.3 EXPERIMENT ON REAL-WORLD DATASETS

To examine the feasibility and effectiveness of our forecasting-based framework, we compare the decoder-only transformer (DOT) with a series of time-series causal discovery methods, including constraint-based and granger causality based methods, on the CausalTime dataset (Cheng et al., 2023). The CausalTime benchmark includes three datasets and corresponding prior graphs in real-world air quality, traffic, and medical scenarios. Since there are only summary graphs annotated, we aggregate the computed relevance and directly compute AUROC and AUPRC based on the summary graph, without binarizing them into discrete causal graphs. Recognizing the data efficiency challenge in the forecasting-to-discovery framework of decoder-only transformers, we employ a low model complexity version (one head, one layer, use a vector instead of tokens as input format for multi-variate) and a sparsity constraint on the gradients to help learn causal graphs more efficiently. We also include a simple two-layer MLP as an alternative predictor, following the same gradient-based structure extraction procedure. Similar to our findings with CNN as model alternative, the

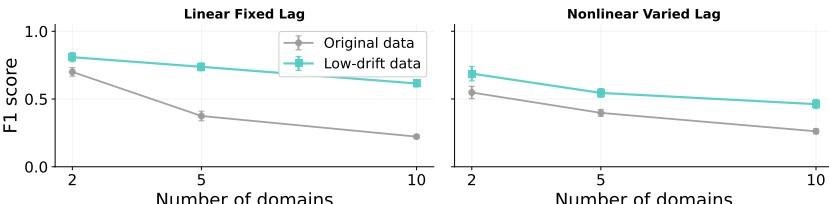

Figure 12: **Performance under non-stationary settings with minimal changes across regimes.** Comparison of F1 scores between randomly sampled regime changes (original data) and minimal changes (low-drift data) where only a small part of the structure is rewired. The minimal changes setting is more realistic and our approach shows much higher data efficiency in both linear and nonlinear non-stationary scenarios, demonstrating the applicability of our approach in real-world settings.

Table 1: Benchmark results on three datasets. The best results are **bolded**, the second-best are underlined, and the third-best are marked with a dagger ($^{\dagger}$). We take the baseline results reported in (Zhou et al., 2024). The result is reported based on five runs with different random seeds.

| Methods | AQI | | Traffic | | Medical | |
|---|---|---|---|---|---|---|
| | AUROC | AUPRC | AUROC | AUPRC | AUROC | AUPRC |
| GC | $0.4538 \pm 0.0377$ | $0.6347 \pm 0.0158$ | $0.4191 \pm 0.0310$ | $0.2789 \pm 0.0018$ | $0.5737 \pm 0.0338$ | $0.4213 \pm 0.0281$ |
| SVAR | $0.6225 \pm 0.0406$ | $0.7903 \pm 0.0175^{\dagger}$ | $0.6329 \pm 0.0047$ | $0.5845 \pm 0.0021$ | $0.7130 \pm 0.0188$ | $0.6774 \pm 0.0358$ |
| N.NTS | $0.5729 \pm 0.0229$ | $0.7100 \pm 0.0228$ | $0.6329 \pm 0.0335$ | $0.5770 \pm 0.0542$ | $0.5019 \pm 0.0682$ | $0.4567 \pm 0.0162$ |
| PCMCI | $0.5272 \pm 0.0744$ | $0.6734 \pm 0.0372$ | $0.5422 \pm 0.0737$ | $0.3474 \pm 0.0581$ | $0.6991 \pm 0.0111$ | $0.5082 \pm 0.0177$ |
| Rhino | $0.6700 \pm 0.0983$ | $0.7593 \pm 0.0755$ | $0.6274 \pm 0.0185$ | $0.3772 \pm 0.0093$ | $0.6520 \pm 0.0212$ | $0.4897 \pm 0.0321$ |
| CUTS | $0.6013 \pm 0.0038$ | $0.5096 \pm 0.0362$ | $0.6238 \pm 0.0179$ | $0.1525 \pm 0.0226$ | $0.3739 \pm 0.0297$ | $0.1537 \pm 0.0039$ |
| CUTS+ | $0.8928 \pm 0.0213^{\dagger}$ | $0.7983 \pm 0.0875$ | $0.6175 \pm 0.0752$ | $\mathbf{0.6367 \pm 0.1197}$ | $\underline{0.8202 \pm 0.0173}$ | $0.5481 \pm 0.1349$ |
| NGC | $0.7172 \pm 0.0076$ | $0.7177 \pm 0.0069$ | $0.6032 \pm 0.0056$ | $0.3583 \pm 0.0495$ | $0.5744 \pm 0.0096$ | $0.4637 \pm 0.0121$ |
| NGM | $0.6728 \pm 0.0164$ | $0.4786 \pm 0.0196$ | $0.4660 \pm 0.0144$ | $0.2826 \pm 0.0098$ | $0.5551 \pm 0.0154$ | $0.4697 \pm 0.0166$ |
| LCCM | $0.8565 \pm 0.0653$ | $\mathbf{0.9260 \pm 0.0246}$ | $0.5545 \pm 0.0254$ | $0.5907 \pm 0.0475^{\dagger}$ | $0.8013 \pm 0.0218^{\dagger}$ | $\underline{0.7554 \pm 0.0235}$ |
| eSRU | $0.8229 \pm 0.0317$ | $0.7223 \pm 0.0317$ | $0.5987 \pm 0.0192$ | $0.4886 \pm 0.0338$ | $0.7559 \pm 0.0365$ | $0.7352 \pm 0.0600^{\dagger}$ |
| SCGL | $0.4915 \pm 0.0476$ | $0.3584 \pm 0.0281$ | $0.5927 \pm 0.0553$ | $0.4544 \pm 0.0315$ | $0.5019 \pm 0.0224$ | $0.4833 \pm 0.0185$ |
| TCDF | $0.4148 \pm 0.0207$ | $0.6527 \pm 0.0087$ | $0.5029 \pm 0.0041$ | $0.3637 \pm 0.0048$ | $0.6329 \pm 0.0384$ | $0.5544 \pm 0.0313$ |
| JRNGC-F | $\mathbf{0.9279 \pm 0.0011}$ | $0.7828 \pm 0.0020$ | $\mathbf{0.7294 \pm 0.0046}$ | $\underline{0.5940 \pm 0.0067}$ | $0.7540 \pm 0.0040$ | $0.7261 \pm 0.0016$ |
| DOT (Ours) | $0.8655 \pm 0.0064$ | $0.6990 \pm 0.0104$ | $\underline{0.7025 \pm 0.0317}$ | $0.5703 \pm 0.0210$ | $0.7237 \pm 0.0175$ | $0.6509 \pm 0.0216$ |
| MLP (Ours) | $\underline{0.9133 \pm 0.0022}$ | $0.7523 \pm 0.0047$ | $0.6345 \pm 0.0155^{\dagger}$ | $0.4811 \pm 0.0075$ | $\mathbf{0.8717 \pm 0.0080}$ | $\mathbf{0.8256 \pm 0.0074}$ |

MLP's lower model complexity and reduced parameter uncertainty lead to higher data efficiency in certain real-world datasets where sample sizes are limited, achieving the best performance on the Medical dataset and competitive results on others. Although DOT is targeted at prediction as the first objective rather than structure learning, we see it surpassing most traditional causal discovery methods, especially ranking among the top three in air quality and traffic datasets. This implies the potential of a prediction-first strategy for causal discovery in the real world, based on transformers. By unifying the prediction and structure learning, the objective becomes clearer and easier to optimize.

### A.10.4    THE SENSITIVITY OF K CHOICE IN THE TOP K BINARIZATION

Below, we show the sensitivity of different values of $k$ in graph binarization. We see that the row-wise top-k has a tradeoff between precision and recall as $k$ increases (see Figure 13). Smaller $k$ emphasizes precision more, while larger values capture more potential parents. In the linear cases, where the model has almost learned the perfect structure, the choice of $k$ significantly affects the outcome when using a much smaller or larger $k$. In the nonlinear case, where the model hasn't fully learned the correct structure, the choice of $k$ does not significantly affect the F1 score. Note that we use row-wise top-k binarization as a straightforward selection method for binary graph benchmarking, assuming each variable has an equal number of parents. This can be replaced with other priors to achieve more robust binarization. For example, we may use the global top-k when assuming that the number of variable parents is not equal, which is much more stable with respect to the choice of $k$ (see Figure 14). It is also encouraged to consider visualizing the heatmap of the

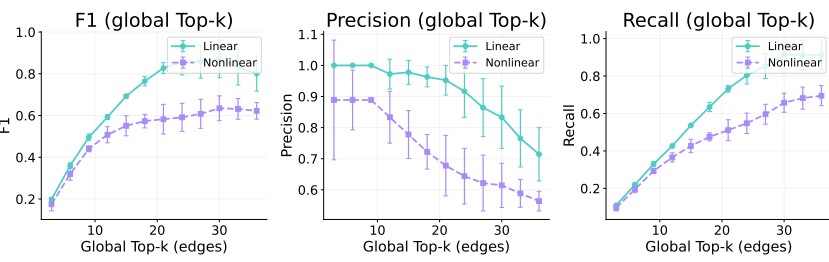

Figure 13: **Sensitivity of $k$ choice in the top-k binarization.** F1 scores of different $k$ choices (1-5) on the linear and nonlinear datasets (the true expectation of incoming degree is 3).

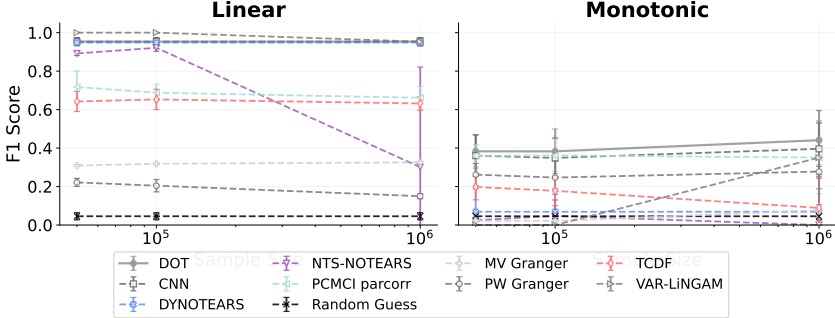

Figure 14: **Sensitivity of $k$ choice in the global top-k binarization.** F1 scores of different global $k$ choices (3-36) on the linear and nonlinear datasets (the true expectation of incoming degree is 3).

edge scores, similar to our uncertainty analysis, to flexibly determine parents based on their needs and interests in practice, depending on the task requirements (precision v.s. recall)

### A.10.5 CONVOLUTIONAL NEURAL NETWORK AS MODEL ALTERNATIVE

Note that our theoretical results for identification is applicable for any architectures, as long as they can perfectly fit the data distribution, which would lead to the structure identification. We use a very simple 1D Convolutional Neural Network as the predictor instead of transformer. It predicts the next time point with the complete lagged window instead of autoregressive learning with teacher forcing.

Figure 15: **Convolutional Neural Network (CNN) as the alternative of transformer.** A 1D CNN is adopted in the same procedure as alternative of transformer (without teacher forcing, it only predicts the next token given full lagged window history).

We see it performs comparably to the deep transformer for most time (see Figure 15). It reminds us the age before Vision Transformer, ResNet often performs best with smaller training data. When data is not sufficient and structural enough (low signal-to-noise ratio), smaller model with stronger inductive bias and lower parameter uncertainty perform higher data efficiency on forecasting, as well as structure learning. We emphasize transformers because its potential of scaling in the long run and the capabilities of handling multi-modal input.

### A.10.6 MORE RESULTS ON UNCERTAINTY ANALYSIS.

Here we show additional results on uncertainty analysis. We use a linear Gaussian dataset and a nonlinear dataset (sigmoid) with 5 variables and 2 lags for the ease of analysis and simplicity of visualization. We take two measures here, original relevance scores and the ranking of relevance scores. The second quantized measure is more stable and comparable across different pairs of variables. The mean over standard deviation of the ranking is introduced as a more general metric, considering the uncertainty and strength of the edges at the same time. We also show two different kinds of top-k for binarization, the row-wise top-k (choose the top-k variables with largest relevance scores for each child variable as its parents, mainly used in the main experiments) and the global top-k (consider the edges with top-k largest relevance scores across all rows/child variables as causal edges). For the row-wise top-k, we take top-3 for each child variable as its parents, and for the global top-k, we take top-15 across all rows as the true edges.

The standard deviation of the original scores is not straightforward to interpret, and we cannot directly compare it across different pairs of variables. This presents an overall trend that larger mean scores are more likely to have larger variance. We use the row-wise ranking of the original relevance scores to compute standard deviation, which is clear and aligns with our intuition that the true causal connections should be more stable than the false ones. We see that the model tends to have a higher consistency for the edges with stronger strengths. These high confidence (low standard deviation and high mean ranking) edges are often the true edges. Based on the ranking, we propose another metric, mean over standard deviation of the ranking, as a more general measure of edge existence. Furthermore, we find for the graph with varied degrees of different nodes, row-wise top-k is a hard truncation and more focused on local structures that might miss and add some edges without any reason. In such circumstances, global top-$k$ is more robust as it considers the most predominant edges in the whole system as true edges. By quantizing the original scores with row-wise rankings, it can also recognize the local structures even when their original strengths are weak. We surprisingly find that the ranking measure is less noisy and gives better results than the original continuous scores, in both linear and nonlinear settings. However, we do not observe any advantages of using the combined metric with global top-$k$ and row-wise top-$k$ compared to only using the mean of rankings, with respect to identification accuracy. We leave the precise calibration and more structure extraction strategies based on uncertainty for future work.

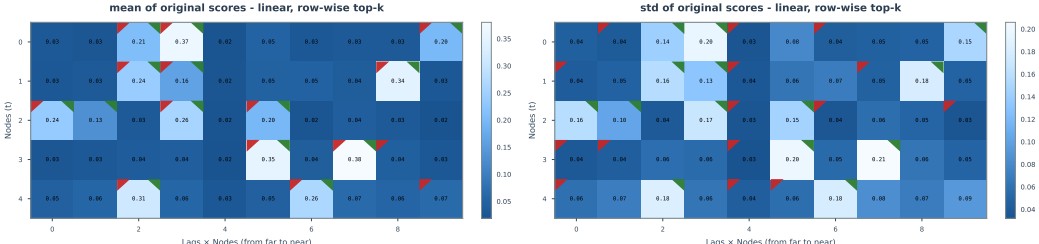

Figure 16: **Mean and standard deviation of relevance scores in the linear setting: (A)** Heatmap showing the mean of edge attributions across all samples. **(B)** Heatmap showing the standard deviation of edge attributions across all samples. The top-left red triangle means that model predicts there is a causal edge and top-right green triangle means that there is a true edge between the two variables.

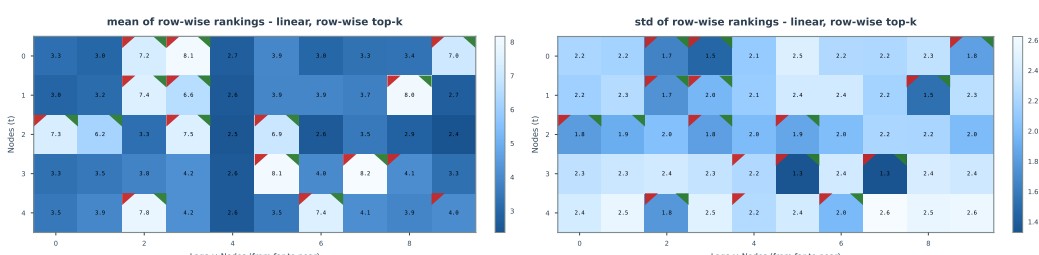

Figure 17: **Mean and standard deviation of row-wise rankings in the linear setting: (A)** Heatmap showing the mean of the ranking of the edge attributions across all samples. **(B)** Heatmap showing the standard deviation of the ranking of the edge attributions across all samples. The top-left red triangle means that model predicts there is a causal edge and top-right green triangle means that there is a true edge between the two variables.

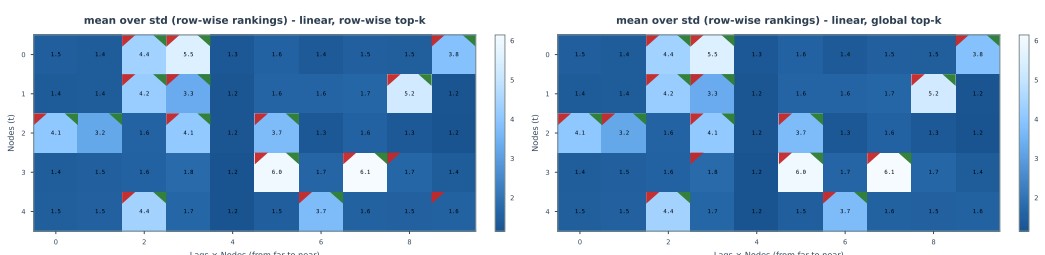

Figure 18: **Row-wise top-k and global top-k in the linear setting: (A)** Heatmap showing the mean over standard deviation of the ranking of the edge attributions across all samples. **(B)** Heatmap showing the standard deviation of the ranking of the edge attributions across all samples. The global top-k select more accurate causal edges than the row-wise top-k. The top-left red triangle means that model predicts there is a causal edge and top-right green triangle means that there is a true edge between the two variables.

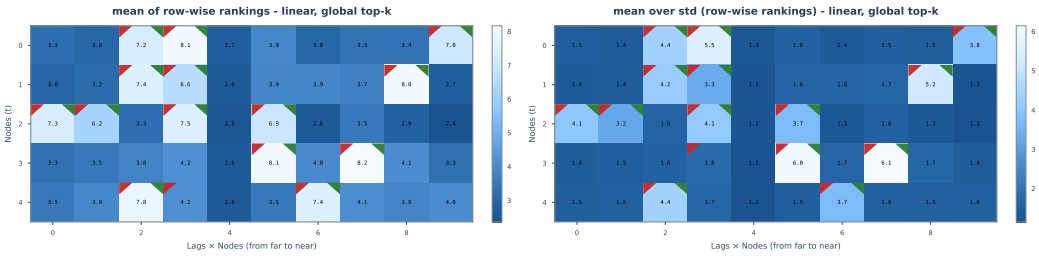

Figure 19: **Global top-k based on mean of rankings and mean over standard deviation of rankings in the linear setting: (A)** Heatmap showing the mean of the ranking of the edge attributions across all samples and predictions selected by the global top-k. **(B)** Heatmap showing the mean over standard deviation of the ranking of the edge attributions across all samples and predictions selected by the global top-k. The top-left red triangle means that model predicts there is a causal edge and top-right green triangle means that there is a true edge between the two variables.

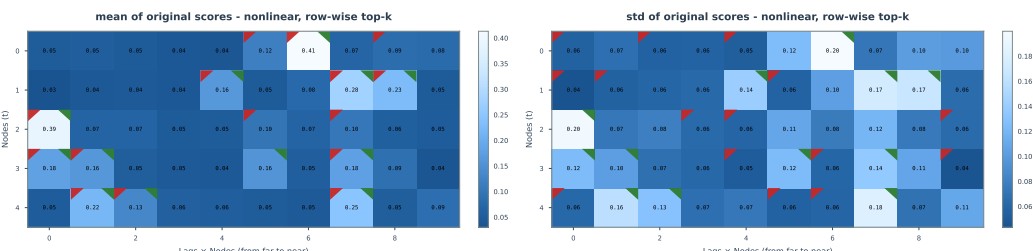

Figure 20: **Mean and standard deviation of relevance scores in the nonlinear setting: (A)** Heatmap showing the mean of edge attributions across all samples. **(B)** Heatmap showing the standard deviation of edge attributions across all samples. The top-left red triangle means that model predicts there is a causal edge and top-right green triangle means that there is a true edge between the two variables.

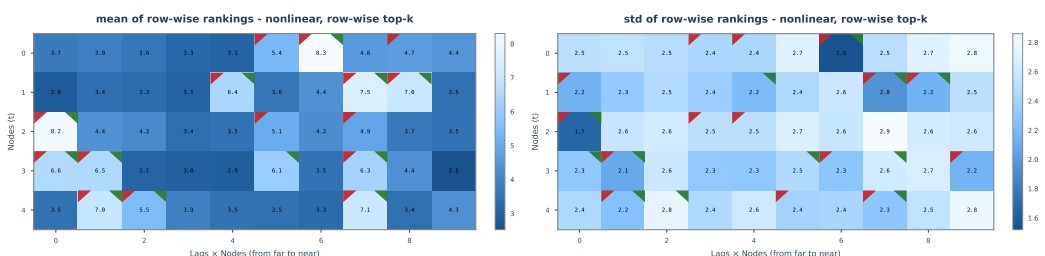

Figure 21: **Mean and standard deviation of row-wise rankings in the nonlinear setting: (A)** Heatmap showing the mean of the ranking of the edge attributions across all samples. **(B)** Heatmap showing the standard deviation of the ranking of the edge attributions across all samples. The top-left red triangle means that model predicts there is a causal edge and top-right green triangle means that there is a true edge between the two variables.

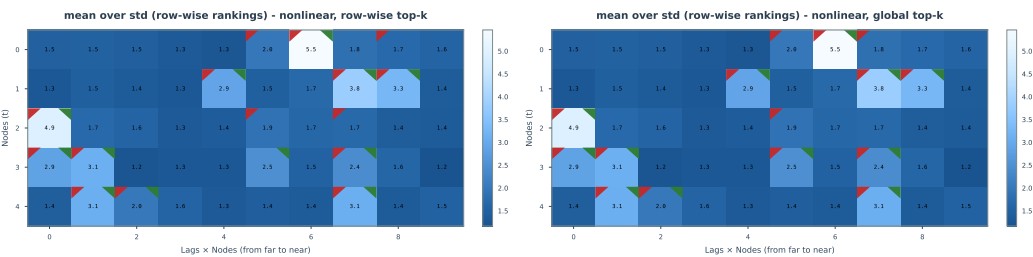

Figure 22: **Row-wise top-k and global top-k in the nonlinear setting: (A)** Heatmap showing the mean over standard deviation of the ranking of the edge attributions across all samples. **(B)** Heatmap showing the standard deviation of the ranking of the edge attributions across all samples. The global top-k select more accurate causal edges than the row-wise top-k. The top-left red triangle means that model predicts there is a causal edge and top-right green triangle means that there is a true edge between the two variables.

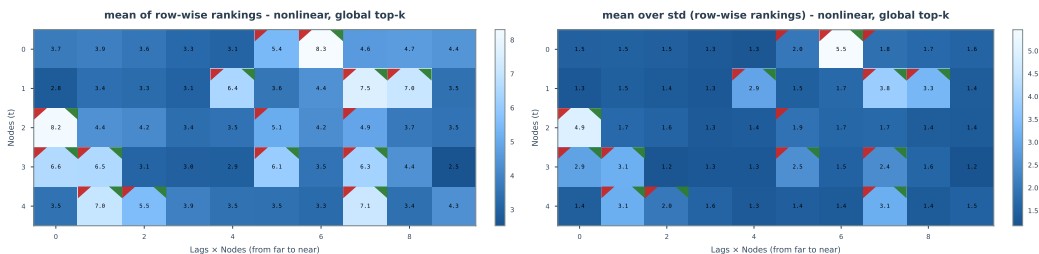

Figure 23: **Global top-k based on mean of rankings and mean over standard deviation of rankings in the nonlinear setting:** **(A)** Heatmap showing the mean of the ranking of the edge attributions across all samples and predictions selected by the global top-k. **(B)** Heatmap showing the mean over standard deviation of the ranking of the edge attributions across all samples and predictions selected by the global top-k. The top-left red triangle means that model predicts there is a causal edge and top-right green triangle means that there is a true edge between the two variables.

We show the histograms of edge strengths in both linear and nonlinear settings. Predictions in both linear and nonlinear settings miss a small portion of edges with low strengths. In nonlinear settings, the prediction strengths are less uniform, and this over-concentration causes it to wrongly identify some edges and miss some true ones. We also find that the ratio of medium-strength edges (0.1 - 0.3) in lag 1 is higher than in the other lags. This shows that the transformer is prone to assign more weights to the last time stamps and pay less attention to the further part, even though they might be true parents. This is largely due to the self-attention mechanism and causal masking, which make the transformer more likely to attend to the last time stamps.

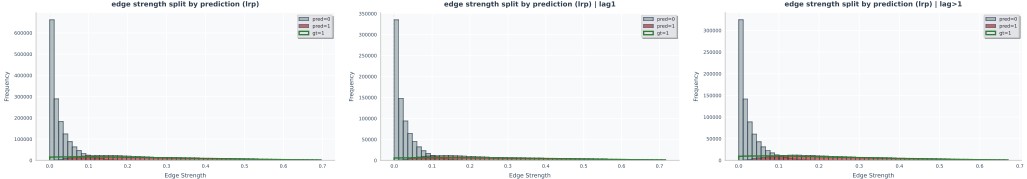

Figure 24: **Histograms of edge strengths in the linear setting:** **(A)** Histogram of edge strengths in the linear setting. **(B)** Histogram of edge strengths in the linear setting with lag 1. **(C)** Histogram of edge strengths in the linear setting with lag larger than 1.

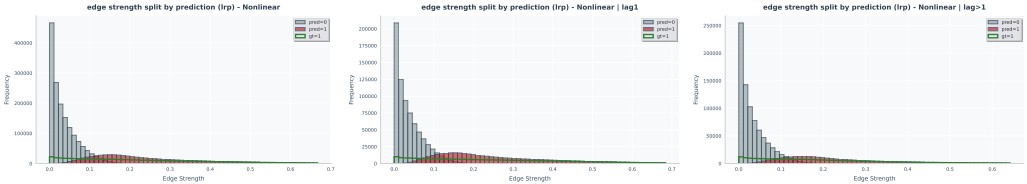

Figure 25: **Histograms of edge strengths in the nonlinear setting:** **(A)** Histogram of edge strengths in the nonlinear setting. **(B)** Histogram of edge strengths in the nonlinear setting with lag 1. **(C)** Histogram of edge strengths in the nonlinear setting with lag larger than 1.

### A.10.7 EXPERIMENT RESULTS BASED ON PRECISION, RECALL AND SHD

Here we show all synthetic experiment results based on precision, recall, F1 score and structural Hamming distance (SHD).

Table 2: High-dimensional (Linear): P/R/F1/SHD (Base: n=10) ('-' indicates results not completed due to numerical instability or timeout.)

| Method | Base | | | | n=25 | | | | n=50 | | | | n=100 | | | |
|---|---|---|---|---|---|---|---|---|---|---|---|---|---|---|---|---|
| | P | R | F1 | SHD | P | R | F1 | SHD | P | R | F1 | SHD | P | R | F1 | SHD |
| DOT | 0.91±0.03 | 1.00±0.00 | 0.95±0.02 | 2.7±1.0 | 0.84±0.03 | 0.98±0.02 | 0.91±0.02 | 12.7±3.1 | 0.84±0.04 | 0.99±0.01 | 0.91±0.02 | 24.3±6.0 | 0.94±0.01 | 1.00±0.00 | 0.97±0.00 | 18.3±2.1 |
| DYNOTEARS | 1.00±0.00 | 0.90±0.02 | 0.95±0.01 | 2.7±0.5 | 1.00±0.00 | 0.85±0.04 | 0.92±0.02 | 10.0±2.6 | 1.00±0.00 | 0.89±0.02 | 0.94±0.01 | 14.0±2.6 | 1.00±0.00 | 0.87±0.01 | 0.93±0.01 | 36.3±3.8 |
| MV Granger | 0.20±0.00 | 0.78±0.09 | 0.32±0.01 | 91.3±7.4 | 0.20±0.00 | 0.75±0.02 | 0.32±0.00 | 209.3±4.0 | 0.20±0.00 | 0.77±0.02 | 0.32±0.00 | 423.7±32.6 | 0.20±0.00 | 0.77±0.01 | 0.32±0.00 | 942.7±2.9 |
| NTS-NOTEARS | 0.78±0.44 | 0.64±0.37 | 0.70±0.40 | 9.7±9.9 | 0.00 | 0.00 | 0.00 | 64.3±2.3 | 0.00 | 0.00 | 0.00 | 127.7±7.1 | 0.33±0.58 | 0.26±0.45 | 0.29±0.50 | 210.3±126.8 |
| PCMCI parcorr | 0.53±0.07 | 1.00±0.00 | 0.69±0.06 | 25.1±6.5 | 0.29±0.02 | 1.00±0.00 | 0.45±0.02 | 160.7±12.4 | 0.17±0.01 | 1.00±0.00 | 0.29±0.01 | 624.7±7.8 | 0.10±0.00 | 1.00±0.00 | 0.18±0.00 | 2500.0±59.6 |
| PW Granger | 0.11±0.02 | 0.88±0.03 | 0.19±0.04 | 211.7±54.9 | 0.08±0.01 | 0.90±0.04 | 0.15±0.02 | 652.0±58.7 | 0.06±0.00 | 0.88±0.02 | 0.11±0.00 | 1840.0±217.9 | 0.05±0.00 | 0.90±0.01 | 0.09±0.00 | 5191.7±215.0 |
| Random Guess | 0.04±0.02 | 0.05±0.02 | 0.05±0.02 | 55.7±3.9 | 0.01±0.02 | 0.01±0.02 | 0.01±0.02 | 125.0±13.5 | 0.01±0.01 | 0.01±0.01 | 0.01±0.01 | 249.0±13.0 | 0.00±0.00 | 0.00±0.00 | 0.00±0.00 | 574.3±32.0 |
| TCDF | 1.00±0.00 | 0.47±0.05 | 0.64±0.04 | 14.3±0.9 | 0.99±0.02 | 0.49±0.03 | 0.65±0.03 | 33.3±1.2 | 0.91±0.02 | 0.62±0.04 | 0.73±0.03 | 57.0±8.7 | 0.99±0.00 | 0.56±0.02 | 0.72±0.02 | 125.0±7.0 |
| VAR-LiNGAM | 0.97±0.05 | 1.00±0.00 | 0.98±0.03 | 0.9±1.5 | 1.00±0.01 | 1.00±0.00 | 1.00±0.00 | 0.3±0.6 | 0.85±0.05 | 1.00±0.00 | 0.92±0.03 | 22.3±7.1 | 0.95±0.01 | 1.00±0.00 | 0.97±0.00 | 16.3±2.5 |

Table 3: High-dimensional (Nonlinear): P/R/F1/SHD (Base: n=10) ('-' indicates results not completed due to numerical instability or timeout.)

| Method | Base | | | | n=25 | | | | n=50 | | | | n=100 | | | |
|---|---|---|---|---|---|---|---|---|---|---|---|---|---|---|---|---|
| | P | R | F1 | SHD | P | R | F1 | SHD | P | R | F1 | SHD | P | R | F1 | SHD |
| DOT | 0.91±0.03 | 1.00±0.00 | 0.95±0.02 | 2.7±1.0 | 0.56±0.01 | 0.65±0.01 | 0.60±0.01 | 55.3±1.2 | 0.54±0.04 | 0.64±0.06 | 0.59±0.05 | 114.3±15.5 | 0.56±0.01 | 0.60±0.01 | 0.58±0.01 | 245.7±8.0 |
| DYNOTEARS | 1.00±0.00 | 0.90±0.02 | 0.95±0.01 | 2.7±0.5 | 1.00±0.00 | 0.05±0.04 | 0.10±0.07 | 61.0±4.6 | 1.00±0.00 | 0.06±0.02 | 0.10±0.04 | 120.7±9.7 | 1.00±0.00 | 0.05±0.00 | 0.09±0.01 | 269.7±2.1 |
| MV Granger | 0.20±0.00 | 0.78±0.09 | 0.32±0.01 | 91.3±7.4 | 0.20±0.00 | 0.04±0.03 | 0.07±0.04 | 72.3±4.9 | 0.19±0.02 | 0.04±0.01 | 0.06±0.01 | 143.3±7.4 | 0.14±0.05 | 0.04±0.01 | 0.06±0.01 | 345.0±23.6 |
| NTS-NOTEARS | 0.78±0.44 | 0.64±0.37 | 0.70±0.40 | 9.7±9.9 | 0.00 | 0.00 | 0.00 | 64.3±2.3 | 0.00 | 0.00 | 0.00 | 127.7±7.1 | 0.33±0.58 | 0.01±0.02 | 0.02±0.03 | 281.0±5.0 |
| PCMCI parcorr | 0.53±0.07 | 1.00±0.00 | 0.69±0.06 | 25.1±6.5 | 0.25±0.02 | 0.82±0.03 | 0.39±0.03 | 169.3±13.1 | 0.14±0.00 | 0.80±0.05 | 0.24±0.01 | 648.0±20.8 | 0.08±0.00 | 0.80±0.00 | 0.15±0.00 | 2571.0±52.0 |
| PW Granger | 0.11±0.02 | 0.88±0.03 | 0.19±0.04 | 211.7±54.9 | 0.16±0.03 | 0.69±0.07 | 0.26±0.04 | 258.0±29.6 | 0.10±0.01 | 0.68±0.05 | 0.18±0.01 | 817.7±80.7 | 0.06±0.00 | 0.68±0.02 | 0.11±0.00 | 2971.3±74.6 |
| Random Guess | 0.04±0.02 | 0.05±0.02 | 0.05±0.02 | 55.7±3.9 | 0.01±0.02 | 0.01±0.02 | 0.01±0.02 | 125.0±13.5 | 0.01±0.01 | 0.01±0.01 | 0.01±0.01 | 249.0±13.0 | 0.00±0.00 | 0.00±0.00 | 0.00±0.00 | 574.3±32.0 |
| TCDF | 1.00±0.00 | 0.47±0.05 | 0.64±0.04 | 14.3±0.9 | 1.00±0.00 | 0.04±0.02 | 0.08±0.05 | 61.7±3.8 | 1.00±0.00 | 0.33±0.05 | 0.50±0.05 | 85.3±10.6 | 1.00±0.00 | 0.10±0.02 | 0.18±0.03 | 255.7±6.7 |
| VAR-LiNGAM | 0.97±0.05 | 1.00±0.00 | 0.98±0.03 | 0.9±1.5 | 0.00 | 0.00 | 0.00 | 77.3±1.5 | 0.00 | 0.00 | 0.00 | 151.7±5.0 | 0.00±0.00 | 0.00±0.00 | 0.00±0.00 | 581.0±3.6 |

Table 4: Long-range Dependencies (Linear): P/R/F1/SHD (Base: k=5) ('-' indicates results not completed due to numerical instability or timeout.)

| Method | Base | | | | k=10 | | | | k=25 | | | | k=50 | | | |
|---|---|---|---|---|---|---|---|---|---|---|---|---|---|---|---|---|
| | P | R | F1 | SHD | P | R | F1 | SHD | P | R | F1 | SHD | P | R | F1 | SHD |
| DOT | 0.91±0.03 | 1.00±0.00 | 0.95±0.02 | 2.7±1.0 | 0.92±0.05 | 1.00±0.00 | 0.96±0.03 | 2.3±1.5 | 0.91±0.05 | 1.00±0.00 | 0.95±0.03 | 2.7±1.5 | 0.96±0.08 | 1.00±0.00 | 0.98±0.04 | 1.3±2.3 |
| DYNOTEARS | 1.00±0.00 | 0.90±0.02 | 0.95±0.01 | 2.7±0.5 | 1.00 | 0.91±0.05 | 0.95±0.03 | 2.7±1.5 | 1.00±0.00 | 0.90±0.08 | 0.95±0.04 | 2.7±2.1 | 1.00±0.00 | 0.80±0.07 | 0.89±0.04 | 5.7±2.1 |
| MV Granger | 0.20±0.00 | 0.78±0.09 | 0.32±0.01 | 91.3±7.4 | 0.10±0.00 | 0.79±0.09 | 0.18±0.00 | 201.0±7.8 | 0.04±0.00 | 0.78±0.07 | 0.08±0.00 | 518.0±48.5 | 0.02±0.00 | 0.72±0.05 | 0.04±0.00 | 1020.7±101.5 |
| NTS-NOTEARS | 0.78±0.44 | 0.64±0.37 | 0.70±0.40 | 9.7±9.9 | 0.00 | 0.00 | 0.00 | 37.3±1.2 | 0.00 | 0.00 | 0.00 | 33.7±0.6 | 0.00 | 0.00 | 0.00 | 33.7±0.6 |
| PCMCI parcorr | 0.53±0.07 | 1.00±0.00 | 0.69±0.06 | 25.1±6.5 | 0.34±0.04 | 1.00±0.00 | 0.51±0.04 | 53.0±6.2 | 0.18±0.02 | 1.00±0.00 | 0.31±0.03 | 123.3±10.3 | 0.10±0.01 | 1.00±0.00 | 0.19±0.02 | 248.7±12.0 |
| PW Granger | 0.11±0.02 | 0.88±0.03 | 0.19±0.04 | 211.7±54.9 | 0.06±0.01 | 0.88±0.04 | 0.12±0.02 | 369.0±53.8 | 0.03±0.00 | 0.88±0.04 | 0.05±0.00 | 897.3±52.0 | 0.01±0.00 | 0.89±0.03 | 0.03±0.00 | 1993.7±263.5 |
| Random Guess | 0.04±0.02 | 0.05±0.02 | 0.05±0.02 | 55.7±3.9 | 0.05±0.02 | 0.05±0.02 | 0.05±0.02 | 54.3±5.5 | 0.00 | 0.00 | 0.00 | 50.3±7.6 | 0.01±0.02 | 0.01±0.02 | 0.01±0.02 | 56.7±2.1 |
| TCDF | 1.00±0.00 | 0.47±0.05 | 0.64±0.04 | 14.3±0.9 | 1.00±0.00 | 0.33±0.09 | 0.49±0.10 | 18.7±3.5 | 1.00±0.00 | 0.22±0.06 | 0.36±0.08 | 21.3±1.2 | 1.00±0.00 | 0.18±0.05 | 0.30±0.07 | 23.7±3.2 |
| VAR-LiNGAM | 0.97±0.05 | 1.00±0.00 | 0.98±0.03 | 0.9±1.5 | 0.92±0.05 | 1.00±0.00 | 0.96±0.03 | 2.3±1.5 | 0.00 | 0.00 | 0.00 | 57.3±1.5 | 0.00 | 0.00 | 0.00 | 58.7±2.3 |

Table 5: Long-range Dependencies (Nonlinear): P/R/F1/SHD (Base: k=5) ('-' indicates results not completed due to numerical instability or timeout.)

| Method | Base | | | | k=10 | | | | k=25 | | | | k=50 | | | |
|---|---|---|---|---|---|---|---|---|---|---|---|---|---|---|---|---|
| | P | R | F1 | SHD | P | R | F1 | SHD | P | R | F1 | SHD | P | R | F1 | SHD |
| DOT | 0.91±0.03 | 1.00±0.00 | 0.95±0.02 | 2.7±1.0 | 0.91±0.04 | 0.99±0.02 | 0.95±0.02 | 3.0±1.0 | 0.91±0.05 | 1.00±0.00 | 0.95±0.03 | 2.7±1.5 | 0.94±0.10 | 0.99±0.02 | 0.96±0.06 | 2.0±3.5 |
| DYNOTEARS | 1.00±0.00 | 0.90±0.02 | 0.95±0.01 | 2.7±0.5 | 1.00 | 0.91±0.05 | 0.95±0.03 | 2.7±1.5 | 1.00±0.00 | 0.90±0.08 | 0.95±0.04 | 2.7±2.1 | 1.00±0.00 | 0.80±0.07 | 0.89±0.04 | 5.7±2.1 |
| MV Granger | 0.20±0.00 | 0.78±0.09 | 0.32±0.01 | 91.3±7.4 | 0.10±0.00 | 0.79±0.09 | 0.18±0.00 | 201.0±7.8 | 0.04±0.00 | 0.78±0.07 | 0.08±0.00 | 518.0±48.5 | 0.02±0.00 | 0.72±0.05 | 0.04±0.00 | 1020.7±101.5 |
| NTS-NOTEARS | 0.78±0.44 | 0.64±0.37 | 0.70±0.40 | 9.7±9.9 | 0.00 | 0.00 | 0.00 | 30.7±3.8 | 0.00 | 0.00 | 0.00 | 29.0±3.0 | 0.00 | 0.00 | 0.00 | 31.3±2.3 |
| PCMCI parcorr | 0.53±0.07 | 1.00±0.00 | 0.69±0.06 | 25.1±6.5 | 0.34±0.04 | 1.00±0.00 | 0.51±0.04 | 53.0±6.2 | 0.18±0.02 | 1.00±0.00 | 0.31±0.03 | 123.3±10.3 | 0.10±0.01 | 1.00±0.00 | 0.19±0.02 | 248.7±12.0 |
| PW Granger | 0.11±0.02 | 0.88±0.03 | 0.19±0.04 | 211.7±54.9 | 0.06±0.01 | 0.88±0.04 | 0.12±0.02 | 369.0±53.8 | 0.03±0.00 | 0.88±0.04 | 0.05±0.00 | 897.3±52.0 | 0.01±0.00 | 0.89±0.03 | 0.03±0.00 | 1993.7±263.5 |
| Random Guess | 0.04±0.02 | 0.05±0.02 | 0.05±0.02 | 55.7±3.9 | 0.05±0.02 | 0.05±0.02 | 0.05±0.02 | 54.3±5.5 | 0.00 | 0.00 | 0.00 | 50.3±7.6 | 0.01±0.02 | 0.01±0.02 | 0.01±0.02 | 56.7±2.1 |
| TCDF | 1.00±0.00 | 0.47±0.05 | 0.64±0.04 | 14.3±0.9 | 1.00±0.00 | 0.33±0.09 | 0.49±0.10 | 18.7±3.5 | 0.33±0.58 | 0.09±0.15 | 0.14±0.24 | 25.0±4.6 | 1.00±0.00 | 0.18±0.05 | 0.30±0.07 | 23.7±3.2 |
| VAR-LiNGAM | 0.97±0.05 | 1.00±0.00 | 0.98±0.03 | 0.9±1.5 | 0.99±0.02 | 1.00±0.00 | 0.99±0.01 | 0.3±0.6 | 0.00 | 0.00 | 0.00 | 55.7±4.7 | 0.00 | 0.00 | 0.00 | 54.7±4.7 |

Table 6: Nonlinearity: Precision, Recall, F1, and SHD (Linear is baseline) ('-' indicates results not completed due to numerical instability or timeout.)

| Method | Linear | | | | CAM-M | | | | CAM-PL+M | | | | MLP-Add | | | | MLP-Cat | | | |
|---|---|---|---|---|---|---|---|---|---|---|---|---|---|---|---|---|---|---|---|---|
| | P | R | F1 | SHD | P | R | F1 | SHD | P | R | F1 | SHD | P | R | F1 | SHD | P | R | F1 | SHD |
| DOT | 0.91±0.03 | 1.00±0.00 | 0.95±0.02 | 2.7±1.0 | 0.39±0.09 | 0.42±0.09 | 0.40±0.09 | 34.2±5.0 | 0.70±0.17 | 0.77±0.17 | 0.73±0.17 | 53.7±77.5 | 0.61±0.12 | 0.67±0.13 | 0.64±0.12 | 20.9±6.9 | 0.55±0.12 | 0.61±0.14 | 0.58±0.13 | 24.2±7.4 |
| DYNOTEARS | 1.00±0.00 | 0.90±0.02 | 0.95±0.01 | 2.7±0.5 | 0.33±0.50 | 0.04±0.06 | 0.07±0.10 | 26.3±2.5 | 0.67±0.48 | 0.31±0.41 | 0.34±0.43 | 60.5±83.5 | 0.00 | 0.00 | 0.00 | 27.3±1.0 | 1.00±0.00 | 0.07±0.08 | 0.09±0.07 | 26.1±0.9 |
| MV Granger | 0.20±0.00 | 0.78±0.09 | 0.32±0.01 | 91.3±7.4 | 0.09±0.11 | 0.03±0.05 | 0.04±0.06 | 29.7±3.9 | 0.10±0.08 | 0.29±0.35 | 0.07±0.06 | 266.1±316.8 | 0.04±0.09 | 0.01±0.03 | 0.01±0.03 | 28.0±1.5 | 0.16±0.09 | 0.07±0.08 | 0.09±0.07 | 33.3±6.0 |
| NTS-NOTEARS | 0.78±0.44 | 0.64±0.37 | 0.70±0.40 | 9.7±9.9 | 0.22±0.44 | 0.01±0.03 | 0.02±0.05 | 27.0±1.6 | 0.04±0.19 | 0.00±0.01 | 0.00±0.01 | 71.8±81.9 | 0.00 | 0.00 | 0.00 | 27.3±1.0 | 0.56±0.53 | 0.02±0.02 | 0.04±0.04 | 26.8±1.2 |
| PCMCI parcorr | 0.53±0.07 | 1.00±0.00 | 0.69±0.06 | 25.1±6.5 | 0.31±0.05 | 0.43±0.09 | 0.36±0.05 | 41.7±5.5 | 0.28±0.16 | 0.88±0.10 | 0.40±0.18 | 433.8±793.0 | 0.48±0.05 | 0.78±0.13 | 0.59±0.06 | 29.4±3.6 | 0.51±0.03 | 0.80±0.13 | 0.62±0.06 | 26.3±3.0 |
| PW Granger | 0.11±0.02 | 0.88±0.03 | 0.19±0.04 | 211.7±54.9 | 0.21±0.05 | 0.35±0.09 | 0.26±0.06 | 53.9±10.3 | 0.13±0.10 | 0.75±0.11 | 0.20±0.17 | 833.4±978.3 | 0.83±0.07 | 0.64±0.12 | 0.77±0.02 | 60.3±9.3 | 0.26±0.04 | 0.70±0.12 | 0.38±0.04 | 64.6±14.1 |
| Random Guess | 0.04±0.02 | 0.05±0.02 | 0.05±0.02 | 55.7±3.9 | 0.04±0.02 | 0.05±0.02 | 0.05±0.02 | 55.7±3.9 | 0.02±0.02 | 0.03±0.03 | 0.02±0.02 | 141.9±168.3 | 0.04±0.02 | 0.05±0.02 | 0.05±0.02 | 55.7±3.9 | 0.04±0.02 | 0.05±0.02 | 0.05±0.02 | 55.7±3.9 |
| TCDF | 1.00±0.00 | 0.47±0.05 | 0.64±0.04 | 14.3±0.9 | 0.78±0.44 | 0.09±0.06 | 0.15±0.11 | 25.0±2.4 | 0.81±0.40 | 0.14±0.13 | 0.23±0.18 | 60.7±73.6 | 0.22±0.44 | 0.01±0.02 | 0.02±0.03 | 27.1±0.9 | 0.67±0.50 | 0.06±0.05 | 0.11±0.08 | 25.8±1.8 |
| VAR-LiNGAM | 0.97±0.05 | 1.00±0.00 | 0.98±0.03 | 0.9±1.5 | 0.11±0.20 | 0.12±0.23 | 0.12±0.21 | 32.9±8.0 | 0.20±0.38 | 0.21±0.40 | 0.20±0.39 | 110.6±174.8 | 0.18±0.35 | 0.20±0.39 | 0.19±0.37 | 29.6±14.7 | 0.18±0.34 | 0.20±0.38 | 0.19±0.36 | 31.0±14.3 |

Table 7: Latent Variables: Precision, Recall, F1, and SHD (Base: 0 latent) ('-' indicates results not completed due to numerical instability or timeout.)

| Method | Base | | | | 3 Latent | | | | 5 Latent | | | | 10 Latent | | | |
|---|---|---|---|---|---|---|---|---|---|---|---|---|---|---|---|---|
| | P | R | F1 | SHD | P | R | F1 | SHD | P | R | F1 | SHD | P | R | F1 | SHD |
| DOT | 0.91±0.03 | 1.00±0.00 | 0.95±0.02 | 2.7±1.0 | 0.76±0.04 | 1.00±0.00 | 0.86±0.02 | 7.3±1.2 | 0.67±0.09 | 1.00±0.00 | 0.80±0.07 | 10.0±2.6 | 0.44±0.18 | 1.00±0.00 | 0.60±0.18 | 16.7±5.5 |
| DYNOTEARS | 1.00±0.00 | 0.90±0.02 | 0.95±0.01 | 2.7±0.5 | 1.00±0.00 | 0.87±0.11 | 0.93±0.06 | 3.0±2.6 | 1.00±0.00 | 0.91±0.09 | 0.95±0.05 | 1.7±1.5 | 1.00±0.00 | 0.87±0.02 | 0.93±0.01 | 1.7±0.6 |
| MV Granger | 0.20±0.00 | 0.78±0.09 | 0.32±0.01 | 91.3±7.4 | 0.20±0.00 | 0.72±0.12 | 0.31±0.01 | 71.7±5.1 | 0.20±0.00 | 0.72±0.12 | 0.31±0.01 | 64.0±14.7 | 0.20±0.00 | 0.62±0.07 | 0.30±0.01 | 38.3±17.9 |
| NTS-NOTEARS | 0.78±0.44 | 0.64±0.37 | 0.70±0.40 | 9.7±9.9 | 1.00±0.00 | 0.78±0.15 | 0.87±0.09 | 5.0±3.6 | 1.00±0.00 | 0.82±0.10 | 0.90±0.07 | 3.3±1.5 | 1.00±0.00 | 0.85±0.07 | 0.92±0.04 | 2.0±1.0 |
| PCMCI parcorr | 0.53±0.07 | 1.00±0.00 | 0.69±0.06 | 25.1±6.5 | 0.42±0.03 | 1.00±0.00 | 0.60±0.03 | 31.0±5.6 | 0.37±0.09 | 1.00±0.00 | 0.54±0.10 | 35.0±10.6 | 0.21±0.12 | 1.00±0.00 | 0.34±0.16 | 52.7±15.7 |
| PW Granger | 0.11±0.02 | 0.88±0.03 | 0.19±0.04 | 211.7±54.9 | 0.12±0.03 | 0.85±0.05 | 0.21±0.05 | 154.7±45.0 | 0.09±0.01 | 0.83±0.07 | 0.16±0.02 | 174.3±31.5 | 0.06±0.02 | 0.74±0.05 | 0.12±0.04 | 146.7±19.6 |
| Random Guess | 0.04±0.02 | 0.05±0.02 | 0.05±0.02 | 55.7±3.9 | 0.05±0.05 | 0.04±0.05 | 0.04±0.05 | 45.7±6.7 | 0.05±0.05 | 0.05±0.05 | 0.05±0.05 | 39.7±3.1 | 0.02±0.03 | 0.02±0.03 | 0.02±0.03 | 24.3±11.9 |
| TCDF | 1.00±0.00 | 0.47±0.05 | 0.64±0.04 | 14.3±0.9 | 0.94±0.05 | 0.46±0.00 | 0.61±0.01 | 13.0±1.0 | 1.00±0.00 | 0.59±0.10 | 0.74±0.08 | 8.3±3.1 | 1.00±0.00 | 0.69±0.16 | 0.81±0.11 | 4.7±3.5 |
| VAR-LiNGAM | 0.97±0.05 | 1.00±0.00 | 0.98±0.03 | 0.9±1.5 | 0.77±0.06 | 1.00±0.00 | 0.87±0.04 | 7.0±2.6 | 0.66±0.17 | 1.00±0.00 | 0.79±0.13 | 11.0±7.0 | 0.35±0.40 | 0.67±0.58 | 0.43±0.44 | 30.3±22.3 |

Table 8: Noise Robustness: Precision, Recall, F1, and SHD (Gaussian is baseline) ('-' indicates results not completed due to numerical instability or timeout.)

| Method | Gaussian | | | | Uniform | | | | Mixed | | | | Hetero-S | | | | Hetero-L | | | |
|---|---|---|---|---|---|---|---|---|---|---|---|---|---|---|---|---|---|---|---|---|
| | P | R | F1 | SHD | P | R | F1 | SHD | P | R | F1 | SHD | P | R | F1 | SHD | P | R | F1 | SHD |
| DOT | 0.91±0.03 | 1.00±0.00 | 0.95±0.02 | 2.7±1.0 | 0.91±0.04 | 1.00±0.00 | 0.95±0.02 | 2.7±1.2 | 0.91±0.04 | 1.00±0.00 | 0.95±0.02 | 2.7±1.2 | 0.91±0.04 | 1.00±0.00 | 0.95±0.02 | 2.7±1.2 | 0.91±0.04 | 1.00±0.00 | 0.95±0.02 | 2.7±1.2 |
| DYNOTEARS | 1.00±0.00 | 0.90±0.02 | 0.95±0.01 | 2.7±0.5 | 1.00±0.00 | 0.91±0.02 | 0.96±0.01 | 2.3±0.6 | 1.00±0.00 | 0.90±0.02 | 0.95±0.01 | 2.7±0.6 | 1.00±0.00 | 0.83±0.07 | 0.91±0.04 | 4.7±2.1 | 1.00±0.00 | 0.79±0.08 | 0.88±0.05 | 5.7±2.3 |
| MV Granger | 0.20±0.00 | 0.78±0.09 | 0.32±0.01 | 91.3±7.4 | 0.20±0.00 | 0.79±0.05 | 0.32±0.00 | 92.3±1.5 | 0.20±0.00 | 0.78±0.08 | 0.32±0.01 | 91.3±2.5 | 0.20±0.00 | 0.83±0.01 | 0.91±0.01 | 4.7±0.6 | 0.20±0.00 | 0.66±0.06 | 0.31±0.01 | 81.3±2.1 |
| NTS-NOTEARS | 0.78±0.44 | 0.64±0.37 | 0.70±0.40 | 9.7±9.9 | 1.00±0.00 | 0.85±0.03 | 0.92±0.02 | 4.0±1.0 | 1.00±0.00 | 0.83±0.01 | 0.91±0.01 | 4.7±0.6 | 1.00±0.00 | 0.77±0.05 | 0.87±0.03 | 6.3±1.5 | 1.00±0.00 | 0.72±0.08 | 0.84±0.06 | 7.7±2.5 |
| PCMCI parcorr | 0.53±0.07 | 1.00±0.00 | 0.69±0.06 | 25.1±6.5 | 0.57±0.10 | 1.00±0.00 | 0.72±0.08 | 21.3±7.5 | 0.51±0.01 | 1.00±0.00 | 0.67±0.01 | 26.3±0.6 | 0.52±0.01 | 1.00±0.00 | 0.69±0.01 | 25.0±1.7 | 0.53±0.01 | 1.00±0.00 | 0.69±0.01 | 24.7±2.3 |
| PW Granger | 0.11±0.02 | 0.88±0.03 | 0.19±0.04 | 211.7±54.9 | 0.04±0.02 | 0.05±0.02 | 0.05±0.02 | 55.7±4.5 | 0.04±0.02 | 0.05±0.02 | 0.05±0.02 | 55.7±4.5 | 0.04±0.02 | 0.05±0.02 | 0.05±0.02 | 55.7±4.5 | 0.07±0.04 | 0.74±0.04 | 0.13±0.02 | 207.0±39.9 |
| Random Guess | 0.04±0.02 | 0.05±0.02 | 0.05±0.02 | 55.7±3.9 | 0.04±0.02 | 0.05±0.02 | 0.05±0.02 | 55.7±4.5 | 0.04±0.02 | 0.05±0.02 | 0.05±0.02 | 55.7±4.5 | 0.04±0.02 | 0.05±0.02 | 0.05±0.02 | 55.7±4.5 | 0.04±0.02 | 0.05±0.02 | 0.05±0.02 | 55.7±4.5 |
| TCDF | 1.00±0.00 | 0.47±0.05 | 0.64±0.04 | 14.3±0.9 | 1.00±0.00 | 0.46±0.04 | 0.63±0.04 | 14.7±0.6 | 1.00±0.00 | 0.47±0.04 | 0.64±0.04 | 14.3±0.6 | 1.00±0.00 | 0.44±0.05 | 0.61±0.05 | 15.3±0.6 | 0.98±0.04 | 0.43±0.04 | 0.59±0.03 | 16.0 |
| VAR-LiNGAM | 0.97±0.05 | 1.00±0.00 | 0.98±0.03 | 0.9±1.5 | 1.00±0.00 | 1.00±0.00 | 1.00±0.00 | 0.0 | 1.00±0.00 | 1.00±0.00 | 1.00±0.00 | 0.0 | 1.00±0.00 | 1.00±0.00 | 1.00±0.00 | 0.0 | 1.00±0.00 | 1.00±0.00 | 1.00±0.00 | 0.0 |

Table 9: Non-stationary (Fixed Lag (k=5), Linear): 2 Domains (Base: stationary) ('-' indicates results not completed due to numerical instability or timeout.)

| Method | Base | | | | 50K | | | | 100K | | | | 1M | | | |
|---|---|---|---|---|---|---|---|---|---|---|---|---|---|---|---|---|
| | P | R | F1 | SHD | P | R | F1 | SHD | P | R | F1 | SHD | P | R | F1 | SHD |
| DOT | 0.91±0.03 | 1.00±0.00 | 0.95±0.02 | 2.7±1.0 | 0.59±0.07 | 0.66±0.12 | 0.62±0.09 | 21.3±5.7 | 0.65±0.06 | 0.73±0.09 | 0.69±0.07 | 17.7±4.0 | 0.75±0.03 | 0.84±0.02 | 0.79±0.01 | 11.7±0.6 |
| DYNOTEARS | 1.00±0.00 | 0.90±0.02 | 0.95±0.01 | 2.7±0.5 | 0.56±0.02 | 0.39±0.12 | 0.45±0.08 | 24.7±1.5 | 0.56±0.02 | 0.39±0.12 | 0.45±0.08 | 24.7±1.5 | 0.56±0.02 | 0.38±0.15 | 0.44±0.10 | 24.7±1.5 |
| MV Granger | 0.20±0.00 | 0.78±0.09 | 0.32±0.01 | 91.3±7.4 | 0.15±0.02 | 0.23±0.05 | 0.17±0.01 | 56.3±9.7 | 0.14±0.01 | 0.45±0.08 | 0.21±0.01 | 92.7±22.1 | - | - | - | - |
| NTS-NOTEARS | 0.78±0.44 | 0.64±0.37 | 0.70±0.40 | 9.7±9.9 | 0.57±0.02 | 0.27±0.11 | 0.35±0.11 | 25.0±2.0 | 0.38±0.33 | 0.21±0.18 | 0.27±0.23 | 25.3±2.5 | 0.00 | 0.00 | 0.00 | 26.7±1.5 |
| PCMCI parcorr | 0.53±0.07 | 1.00±0.00 | 0.69±0.06 | 25.1±6.5 | 0.17±0.02 | 0.99±0.02 | 0.29±0.03 | 131.3±9.3 | 0.13±0.00 | 0.99±0.02 | 0.24±0.01 | 171.3±11.1 | 0.08±0.00 | 1.00±0.00 | 0.15±0.00 | 303.7±16.4 |
| PW Granger | 0.11±0.02 | 0.88±0.03 | 0.19±0.04 | 211.7±54.9 | 0.09±0.01 | 0.86±0.01 | 0.16±0.01 | 243.7±11.0 | 0.08±0.00 | 0.86±0.01 | 0.14±0.00 | 276.0±16.5 | 0.06±0.00 | 0.86±0.01 | 0.11±0.00 | 362.3±25.4 |
| Random Guess | 0.04±0.02 | 0.05±0.02 | 0.05±0.02 | 55.7±3.9 | 0.05±0.02 | 0.06±0.03 | 0.06±0.03 | 54.3±5.5 | 0.05±0.02 | 0.06±0.03 | 0.06±0.03 | 54.3±5.5 | 0.05±0.02 | 0.06±0.03 | 0.06±0.03 | 54.3±5.5 |
| TCDF | 1.00±0.00 | 0.47±0.05 | 0.64±0.04 | 14.3±0.9 | 0.55±0.04 | 0.30±0.06 | 0.39±0.05 | 25.3±2.5 | 0.56±0.02 | 0.32±0.05 | 0.40±0.04 | 25.0±2.0 | 0.00 | 0.00 | 0.00 | 26.7±1.5 |
| VAR-LiNGAM | 0.97±0.05 | 1.00±0.00 | 0.98±0.03 | 0.9±1.5 | 0.29±0.06 | 0.99±0.02 | 0.44±0.08 | 68.3±17.9 | 0.23±0.02 | 0.99±0.02 | 0.37±0.02 | 90.3±3.8 | 0.53±0.01 | 0.59±0.03 | 0.56±0.01 | 25.0±1.0 |

Table 10: Non-stationary (Fixed Lag (k=5), Linear): 5 Domains (Base: stationary) ('-' indicates results not completed due to numerical instability or timeout.)

| Method | Base | | | | 50K | | | | 100K | | | | 1M | | | |
|---|---|---|---|---|---|---|---|---|---|---|---|---|---|---|---|---|
| | P | R | F1 | SHD | P | R | F1 | SHD | P | R | F1 | SHD | P | R | F1 | SHD |
| DOT | 0.91±0.03 | 1.00±0.00 | 0.95±0.02 | 2.7±1.0 | 0.29±0.01 | 0.31±0.03 | 0.30±0.01 | 41.3±3.5 | 0.32±0.04 | 0.35±0.07 | 0.33±0.05 | 39.3±5.5 | 0.48±0.07 | 0.52±0.11 | 0.50±0.09 | 30.0±7.0 |
| DYNOTEARS | 1.00±0.00 | 0.90±0.02 | 0.95±0.01 | 2.7±0.5 | 0.18±0.31 | 0.02±0.04 | 0.04±0.07 | 28.7±3.5 | 0.18±0.31 | 0.02±0.04 | 0.04±0.07 | 28.7±3.5 | 0.20±0.35 | 0.02±0.03 | 0.03±0.05 | 28.7±3.5 |
| MV Granger | 0.20±0.00 | 0.78±0.09 | 0.32±0.01 | 91.3±7.4 | 0.00 | 0.00 | 0.00 | 28.7±3.5 | 0.03±0.05 | 0.00±0.01 | 0.01±0.01 | 30.0±1.7 | - | - | - | - |
| NTS-NOTEARS | 0.78±0.44 | 0.64±0.37 | 0.70±0.40 | 9.7±9.9 | 0.27±0.46 | 0.01±0.02 | 0.02±0.04 | 28.3±4.0 | 0.27±0.46 | 0.01±0.02 | 0.02±0.04 | 28.3±4.0 | 0.00 | 0.00 | 0.00 | 28.7±3.5 |
| PCMCI parcorr | 0.53±0.07 | 1.00±0.00 | 0.69±0.06 | 25.1±6.5 | 0.13±0.01 | 0.93±0.01 | 0.23±0.01 | 185.0±15.4 | 0.11±0.01 | 0.96±0.01 | 0.20±0.02 | 228.7±8.4 | 0.07±0.01 | 0.99±0.01 | 0.13±0.01 | 367.7±7.1 |
| PW Granger | 0.11±0.02 | 0.88±0.03 | 0.19±0.04 | 211.7±54.9 | 0.08±0.01 | 0.85±0.04 | 0.15±0.02 | 276.3±10.3 | 0.07±0.01 | 0.87±0.03 | 0.14±0.02 | 318.7±5.8 | 0.06±0.01 | 0.88±0.03 | 0.11±0.02 | 397.0±10.6 |
| Random Guess | 0.04±0.02 | 0.05±0.02 | 0.05±0.02 | 55.7±3.9 | 0.06±0.01 | 0.06±0.01 | 0.06±0.01 | 56.0±6.9 | 0.06±0.01 | 0.06±0.01 | 0.06±0.01 | 56.0±6.9 | 0.06±0.01 | 0.06±0.01 | 0.06±0.01 | 56.0±6.9 |
| TCDF | 1.00±0.00 | 0.47±0.05 | 0.64±0.04 | 14.3±0.9 | 0.49±0.10 | 0.04±0.01 | 0.08±0.02 | 28.7±3.5 | 0.46±0.12 | 0.05±0.05 | 0.09±0.07 | 29.3±2.5 | 0.00 | 0.00 | 0.00 | 28.7±3.5 |
| VAR-LiNGAM | 0.97±0.05 | 1.00±0.00 | 0.98±0.03 | 0.9±1.5 | 0.16±0.00 | 0.89±0.01 | 0.28±0.01 | 133.7±17.0 | 0.15±0.00 | 0.94±0.01 | 0.25±0.00 | 161.7±18.5 | 0.26±0.00 | 0.28±0.02 | 0.27±0.01 | 42.7±3.5 |

Table 11: Non-stationary (Fixed Lag (k=5), Linear): 10 Domains (Base: stationary) ('-' indicates results not completed due to numerical instability or timeout.)

| Method | Base | | | | 50K | | | | 100K | | | | 1M | | | |
|---|---|---|---|---|---|---|---|---|---|---|---|---|---|---|---|---|
| | P | R | F1 | SHD | P | R | F1 | SHD | P | R | F1 | SHD | P | R | F1 | SHD |
| DOT | 0.91±0.03 | 1.00±0.00 | 0.95±0.02 | 2.7±1.0 | 0.19±0.03 | 0.21±0.03 | 0.20±0.03 | 46.0±2.6 | 0.21±0.03 | 0.23±0.03 | 0.21±0.03 | 45.0±2.6 | 0.24±0.02 | 0.27±0.02 | 0.25±0.02 | 43.0±1.0 |
| DYNOTEARS | 1.00±0.00 | 0.90±0.02 | 0.95±0.01 | 2.7±0.5 | 0.00 | 0.00 | 0.00 | 27.7±1.5 | 0.00 | 0.00 | 0.00 | 27.7±1.5 | 0.00 | 0.00 | 0.00 | 27.7±1.5 |
| MV Granger | 0.20±0.00 | 0.78±0.09 | 0.32±0.01 | 91.3±7.4 | 0.00 | 0.00 | 0.00 | 27.7±1.5 | 0.00 | 0.00 | 0.00 | 27.7±1.5 | - | - | - | - |
| NTS-NOTEARS | 0.78±0.44 | 0.64±0.37 | 0.70±0.40 | 9.7±9.9 | 0.00 | 0.00 | 0.00 | 27.7±1.5 | 0.00 | 0.00 | 0.00 | 27.7±1.5 | 0.00 | 0.00 | 0.00 | 27.7±1.5 |
| PCMCI parcorr | 0.53±0.07 | 1.00±0.00 | 0.69±0.06 | 25.1±6.5 | 0.10±0.01 | 0.78±0.03 | 0.17±0.01 | 204.3±13.7 | 0.09±0.00 | 0.86±0.02 | 0.16±0.00 | 249.7±8.1 | 0.07 | 0.96 | 0.12 | 382.0 |
| PW Granger | 0.11±0.02 | 0.88±0.03 | 0.19±0.04 | 211.7±54.9 | 0.07±0.00 | 0.79±0.05 | 0.13±0.01 | 305.0±17.8 | 0.07±0.00 | 0.86±0.03 | 0.12±0.01 | 346.3±4.7 | 0.06±0.00 | 0.90±0.02 | 0.11±0.01 | 403.7±4.7 |
| Random Guess | 0.04±0.02 | 0.05±0.02 | 0.05±0.02 | 55.7±3.9 | 0.05±0.01 | 0.06±0.01 | 0.06±0.01 | 55.0±5.2 | 0.06±0.01 | 0.06±0.01 | 0.06±0.01 | 55.0±5.2 | 0.05±0.01 | 0.06±0.01 | 0.06±0.01 | 55.0±5.2 |
| TCDF | 1.00±0.00 | 0.47±0.05 | 0.64±0.04 | 14.3±0.9 | 0.15±0.26 | 0.01±0.02 | 0.02±0.04 | 27.7±1.5 | 0.25±0.23 | 0.02±0.02 | 0.03±0.03 | 27.7±1.5 | 0.00 | 0.00 | 0.00 | 27.7±1.5 |
| VAR-LiNGAM | 0.97±0.05 | 1.00±0.00 | 0.98±0.03 | 0.9±1.5 | 0.05±0.01 | 0.16±0.09 | 0.07±0.02 | 113.0±39.2 | 0.10±0.00 | 0.79±0.05 | 0.18±0.00 | 202.7±4.6 | 0.18±0.02 | 0.21±0.03 | 0.19±0.03 | 46.7±2.1 |

Table 12: Non-stationary (Variable Lag (k=1-5), Nonlinear): 2 Domains (Base: stationary) ('-' indicates results not completed due to numerical instability or timeout.)

| Method | Base | | | | 50K | | | | 100K | | | | 1M | | | |
|---|---|---|---|---|---|---|---|---|---|---|---|---|---|---|---|---|
| | P | R | F1 | SHD | P | R | F1 | SHD | P | R | F1 | SHD | P | R | F1 | SHD |
| DOT | 0.91±0.03 | 1.00±0.00 | 0.95±0.02 | 2.7±1.0 | 0.44±0.18 | 0.58±0.10 | 0.50±0.15 | 24.3±5.5 | 0.49±0.18 | 0.66±0.09 | 0.55±0.15 | 21.3±5.5 | 0.53±0.18 | 0.71±0.07 | 0.59±0.15 | 19.3±4.9 |
| DYNOTEARS | 1.00±0.00 | 0.90±0.02 | 0.95±0.01 | 2.7±0.5 | 0.21±0.25 | 0.23±0.16 | 0.20±0.19 | 49.0±43.3 | 0.21±0.25 | 0.22±0.15 | 0.19±0.17 | 50.3±45.6 | 0.21±0.25 | 0.22±0.15 | 0.19±0.17 | 49.3±43.9 |
| MV Granger | 0.20±0.00 | 0.78±0.09 | 0.32±0.01 | 91.3±7.4 | 0.10±0.04 | 0.14±0.08 | 0.11±0.06 | 44.3±20.6 | 0.09±0.03 | 0.20±0.11 | 0.13±0.05 | 59.3±29.6 | - | - | - | - |
| NTS-NOTEARS | 0.78±0.44 | 0.64±0.37 | 0.70±0.40 | 9.7±9.9 | 0.50±0.00 | 0.18±0.12 | 0.25±0.14 | 21.0±6.1 | 0.17±0.29 | 0.08±0.13 | 0.11±0.18 | 21.0±6.4 | 0.00 | 0.00 | 0.00 | 21.0±6.1 |
| PCMCI parcorr | 0.53±0.07 | 1.00±0.00 | 0.69±0.06 | 25.1±6.5 | 0.10±0.01 | 0.74±0.11 | 0.18±0.02 | 150.3±45.4 | 0.09±0.00 | 0.79±0.05 | 0.16±0.00 | 178.3±52.8 | 0.06±0.00 | 0.88±0.04 | 0.11±0.01 | 292.7±72.5 |
| PW Granger | 0.11±0.02 | 0.88±0.03 | 0.19±0.04 | 211.7±54.9 | 0.05±0.01 | 0.73±0.11 | 0.09±0.02 | 300.3±80.6 | 0.05±0.01 | 0.76±0.07 | 0.09±0.02 | 331.3±76.7 | 0.04±0.01 | 0.79±0.06 | 0.08±0.02 | 380.0±50.7 |
| Random Guess | 0.04±0.02 | 0.05±0.02 | 0.05±0.02 | 55.7±3.9 | 0.03±0.03 | 0.03±0.05 | 0.03±0.04 | 40.7±16.2 | 0.03±0.03 | 0.03±0.05 | 0.03±0.04 | 40.7±16.2 | 0.03±0.03 | 0.03±0.05 | 0.03±0.04 | 40.7±16.2 |
| TCDF | 1.00±0.00 | 0.47±0.05 | 0.64±0.04 | 14.3±0.9 | 0.46±0.11 | 0.17±0.05 | 0.25±0.07 | 21.3±4.7 | 0.46±0.11 | 0.17±0.05 | 0.25±0.07 | 21.3±4.7 | 0.00 | 0.00 | 0.00 | 21.0±6.1 |
| VAR-LiNGAM | 0.97±0.05 | 1.00±0.00 | 0.98±0.03 | 0.9±1.5 | 0.05±0.04 | 0.34±0.32 | 0.09±0.06 | 141.0±52.8 | 0.10±0.01 | 0.74±0.07 | 0.17±0.02 | 160.0±51.7 | 0.37±0.13 | 0.47±0.07 | 0.40±0.11 | 29.0±2.6 |

Table 13: Non-stationary (Variable Lag (k=1-5), Nonlinear): 5 Domains (Base: stationary) ('-' indicates results not completed due to numerical instability or timeout.)

| Method | Base | | | | 50K | | | | 100K | | | | 1M | | | |
|---|---|---|---|---|---|---|---|---|---|---|---|---|---|---|---|---|
| | P | R | F1 | SHD | P | R | F1 | SHD | P | R | F1 | SHD | P | R | F1 | SHD |
| DOT | 0.91±0.03 | 1.00±0.00 | 0.95±0.02 | 2.7±1.0 | 0.30±0.04 | 0.38±0.07 | 0.33±0.05 | 35.3±4.0 | 0.35±0.05 | 0.44±0.09 | 0.39±0.06 | 32.0±4.6 | 0.44±0.07 | 0.53±0.05 | 0.47±0.06 | 27.3±2.5 |
| DYNOTEARS | 1.00±0.00 | 0.90±0.02 | 0.95±0.01 | 2.7±0.5 | 0.09±0.06 | 0.13±0.16 | 0.08±0.06 | 58.3±51.7 | 0.09±0.06 | 0.13±0.16 | 0.08±0.06 | 58.7±51.5 | 0.09±0.06 | 0.13±0.16 | 0.08±0.06 | 58.0±51.2 |
| MV Granger | 0.20±0.00 | 0.78±0.09 | 0.32±0.01 | 91.3±7.4 | 0.06±0.05 | 0.02±0.02 | 0.04±0.03 | 29.3±7.2 | 0.08±0.00 | 0.04±0.03 | 0.05±0.02 | 35.0±11.1 | - | - | - | - |
| NTS-NOTEARS | 0.78±0.44 | 0.64±0.37 | 0.70±0.40 | 9.7±9.9 | 0.04±0.08 | 0.01±0.02 | 0.02±0.03 | 25.3±4.5 | 0.06±0.10 | 0.01±0.02 | 0.02±0.04 | 25.3±4.5 | 0.00 | 0.00 | 0.00 | 23.7±2.5 |
| PCMCI parcorr | 0.53±0.07 | 1.00±0.00 | 0.69±0.06 | 25.1±6.5 | 0.13±0.01 | 0.81±0.03 | 0.15±0.02 | 229.7±36.5 | 0.07±0.01 | 0.84±0.02 | 0.13±0.02 | 271.7±41.5 | 0.06±0.00 | 0.95±0.02 | 0.11±0.00 | 396.0±19.8 |
| PW Granger | 0.11±0.02 | 0.88±0.03 | 0.19±0.04 | 211.7±54.9 | 0.05±0.01 | 0.77±0.07 | 0.09±0.01 | 374.7±43.0 | 0.05±0.01 | 0.79±0.07 | 0.09±0.01 | 393.0±30.0 | 0.05±0.01 | 0.84±0.04 | 0.09±0.01 | 426.7±7.5 |
| Random Guess | 0.04±0.02 | 0.05±0.02 | 0.05±0.02 | 55.7±3.9 | 0.06±0.02 | 0.06±0.04 | 0.06±0.03 | 42.0±12.1 | 0.06±0.04 | 0.06±0.04 | 0.06±0.04 | 42.0±12.1 | 0.06±0.02 | 0.06±0.04 | 0.06±0.03 | 42.0±12.1 |
| TCDF | 1.00±0.00 | 0.47±0.05 | 0.64±0.04 | 14.3±0.9 | 0.26±0.03 | 0.05±0.03 | 0.09±0.04 | 26.0±1.7 | 0.28±0.12 | 0.05±0.02 | 0.09±0.03 | 25.7±2.5 | 0.25±0.05 | 0.06±0.04 | 0.09±0.04 | 26.3±2.1 |
| VAR-LiNGAM | 0.97±0.05 | 1.00±0.00 | 0.98±0.03 | 0.9±1.5 | 0.06±0.02 | 0.57±0.30 | 0.11±0.04 | 205.3±57.7 | 0.08±0.02 | 0.77±0.03 | 0.14±0.04 | 247.0±74.1 | 0.22±0.02 | 0.27±0.04 | 0.24±0.03 | 40.0±3.6 |

Table 14: Non-stationary (Variable Lag (k=1-5), Nonlinear): 10 Domains (Base: stationary) ('-' indicates results not completed due to numerical instability or timeout.)

| Method | Base | | | | 50K | | | | 100K | | | | 1M | | | |
|---|---|---|---|---|---|---|---|---|---|---|---|---|---|---|---|---|
| | P | R | F1 | SHD | P | R | F1 | SHD | P | R | F1 | SHD | P | R | F1 | SHD |
| DOT | 0.91±0.03 | 1.00±0.00 | 0.95±0.02 | 2.7±1.0 | 0.20±0.04 | 0.25±0.04 | 0.22±0.04 | 42.7±3.1 | 0.24±0.05 | 0.30±0.05 | 0.26±0.05 | 41.0±2.6 | 0.28±0.04 | 0.35±0.05 | 0.30±0.05 | 38.0±3.0 |
| DYNOTEARS | 1.00±0.00 | 0.90±0.02 | 0.95±0.01 | 2.7±0.5 | 0.04±0.02 | 0.11±0.11 | 0.05±0.04 | 71.0±36.3 | 0.04±0.02 | 0.11±0.12 | 0.06±0.04 | 73.3±37.9 | 0.04±0.02 | 0.11±0.12 | 0.05±0.04 | 71.3±35.9 |
| MV Granger | 0.20±0.00 | 0.78±0.09 | 0.32±0.01 | 91.3±7.4 | 0.06±0.00 | 0.02±0.03 | 0.02±0.02 | 30.3±7.0 | 0.06±0.02 | 0.04±0.03 | 0.04±0.03 | 41.3±11.7 | 0.04 | 0.09 | 0.05 | 86.0 |
| NTS-NOTEARS | 0.78±0.44 | 0.64±0.37 | 0.70±0.40 | 9.7±9.9 | 0.07±0.06 | 0.01±0.01 | 0.02±0.02 | 28.0±2.6 | 0.06±0.05 | 0.01±0.01 | 0.02±0.02 | 28.0±3.0 | 0.00 | 0.00 | 0.00 | 25.0±2.0 |
| PCMCI parcorr | 0.53±0.07 | 1.00±0.00 | 0.69±0.06 | 25.1±6.5 | 0.07±0.01 | 0.71±0.07 | 0.12±0.02 | 256.3±23.3 | 0.06±0.01 | 0.78±0.05 | 0.11±0.02 | 305.0±25.5 | 0.06 | 0.95 | 0.11 | 426.0 |
| PW Granger | 0.11±0.02 | 0.88±0.03 | 0.19±0.04 | 211.7±54.9 | 0.05±0.00 | 0.79±0.04 | 0.09±0.00 | 382.7±24.9 | 0.05±0.00 | 0.82±0.03 | 0.10±0.00 | 398.7±19.1 | 0.05±0.00 | 0.86±0.01 | 0.09±0.01 | 424.0±1.7 |
| Random Guess | 0.04±0.02 | 0.05±0.02 | 0.05±0.02 | 55.7±3.9 | 0.06±0.01 | 0.05±0.04 | 0.05±0.03 | 43.7±8.4 | 0.06±0.01 | 0.05±0.04 | 0.05±0.03 | 43.7±8.4 | 0.06±0.01 | 0.05±0.04 | 0.05±0.03 | 43.7±8.4 |
| TCDF | 1.00±0.00 | 0.47±0.05 | 0.64±0.04 | 14.3±0.9 | 0.10±0.06 | 0.02±0.03 | 0.04±0.04 | 29.0±2.0 | 0.11±0.06 | 0.03±0.02 | 0.04±0.03 | 30.0±2.0 | 0.00 | 0.00 | 0.00 | 25.0±2.0 |
| VAR-LiNGAM | 0.97±0.05 | 1.00±0.00 | 0.98±0.03 | 0.9±1.5 | 0.07±0.01 | 0.67±0.10 | 0.12±0.01 | 246.0±26.7 | 0.06±0.01 | 0.76±0.03 | 0.11±0.01 | 303.3±24.2 | 0.09±0.02 | 0.12±0.03 | 0.10±0.03 | 49.3±1.5 |

