# OpenReview forum: "Transformer Is Inherently a Causal Learner"
_ICLR.cc/2026/Conference — Submitted to ICLR 2026_

### Official Review · Reviewer_bY8W · 2025-10-29

**Soundness:** 3
**Presentation:** 2
**Contribution:** 2
**Rating:** 2
**Confidence:** 5

**Summary:**

This paper aims to provide a theoretical framework that the gradient sensitivities of transformer outputs w.r.t. past inputs directly recover the underlying causal graph.  They provided experimental results to support their conclusions and compared them with many causal discovery baselines.

**Strengths:**

1. The authors proved that decoder-only transformers trained autoregressively admit causal identifiability.
2. They offered clear and plausible theoretical motivation for their DOT algorithm.
3. The authors tested DOT's performance and compared it with previous methods.

**Weaknesses:**

1. The proposed method is not well-validated experimentally. How does the causal graph obtained using DOT compare to the ground truth? Are there any visualizations of the results, as well as more evaluation metrics such as SHD?

2. Some problems arose during the literature review. The causal relationship mentioned in the article is basically based on Granger causality, but DYNOTEARS does not deal with Granger causality, so it should not be discussed in a unified manner.

3. There are already some methods that use gradients for Granger causality detection, such as JNRGC, but DOT does not compare them or fully discuss the differences between them. In fact, JNRGC can also be flexibly applied in the Transformer framework.

4. The LRP attribution method used does not seem to be the author's original invention, but a direct reference to other algorithms. I suspect this is not novel enough.

[1] Zhou, W., Bai, S., Yu, S., Zhao, Q., & Chen, B. (2024). Jacobian regularizer-based neural granger causality. arXiv preprint arXiv:2405.08779.

**Questions:**

1. What is the approximate range of the sample size required for DOT? Is there any relationship between it and the number of nodes and causal structure?

2. Can DOT handle dynamic causal graphs? Is there a specific case study?

3. Figure 5 reports DOT’s ability to cope with noise sequences. Does different noise intensities have a big impact on the algorithm?

4. What is the maximum number of nodes that DOT can handle? Are there any experiments and visualizations on real-world datasets, such as CausalTime?

---

> ### Author Response · Authors · 2025-12-04
> **Response to Reviewer bY8W**
>
> We thank the reviewer for the thorough evaluation and for highlighting important points on experimental validation and related work. We respond to each concern below:
>
> **W1 & Q4: Experimental Validation, Visualizations, Metrics, and CausalTime**
> > W1: How does the causal graph compare to ground truth? Visualizations and SHD metrics?
>
> > Q4: Maximum number of nodes? Experiments on real-world datasets like CausalTime?
>
> We have conducted comprehensive experiments covering high-dimensional, long-range, nonlinear, nonstationary, noise types, and latent variable settings, all evaluated against ground truth causal graphs. We provide:
> - **Visualizations** in the uncertainty analysis section (Appendix A.10.6)
> - **Additional metrics** (precision, recall, SHD) in Appendix A.10.7
> - **Real-world experiments on CausalTime** (Appendix A.10.3), where DOT ranks among the top performers on air quality and traffic datasets
> - **Assumption violation handling** in Figure 6
>
> The maximum number of nodes depends on the input sequence length and transformer context window length.
>
> **W2: Unified Framework vs. Granger Causality**
> > DYNOTEARS does not deal with Granger causality, so it should not be discussed in a unified manner.
>
> We build a unified view of forecasting and causal discovery through transformers as a general solution to time-series causal discovery. We therefore compare different families of algorithms, including but not limited to Granger causality methods.
>
> **W3: Comparison with JRNGC**
> > JRNGC uses gradients for Granger causality detection but is not compared.
>
> Our procedure shares similarities with JRNGC but differs in key aspects:
>
> (1) **Theoretical foundation**: We ground our framework in *structural causality* with formal identification guarantees (Theorem 1: H_j = 0 ⟺ j ∉ S), while JRNGC is based on Granger causality without such results.
>
> (2) **Unified prediction-identification view**: We establish a principled connection between autoregressive forecasting and causal structure learning, enabling foundation model alignment. JRNGC focuses on gradient regularization without this broader perspective.
>
> (3) **Handling assumption violations**: We develop systematic post-processing strategies using L-PCMCI and PCMCI+ (Figure 6), not addressed by JRNGC.
>
> (4) **Architecture alignment**: Our framework specifically connects to decoder-only transformers' properties as scalable causal learners.
>
> We have included JRNGC-F as a baseline in CausalTime experiments (Table 1), where both methods show competitive performance.
>
> **W4: LRP Novelty**
> > LRP is not the author's original invention.
>
> Our contribution lies not in inventing a new attribution method, but in establishing the theoretical connection between gradient-based attributions and structural causal identification (Theorem 1). Given our general result based on gradient energy, any gradient-based attribution method can serve as a causal extraction tool. We adopt LRP for its stability and popularity in LLM interpretability.
>
> **Q1: Sample Size Requirements**
> > What is the approximate range of sample size required? Relationship to nodes and structure?
>
> Required sample size depends on nonlinearity and graph density—the fundamental question is how much data is needed to fit the distribution with a transformer. Adding sparsity regularization should improve data efficiency. See Figure 3 and 4 for sample size effects on graph recovery accuracy.
>
> **Q2: Dynamic Causal Graphs**
> > Can DOT handle dynamic causal graphs?
>
> Yes, this is a key advantage. Our nonstationary environment experiments (Figure 4) demonstrate that DOT natively handles dynamic causal graphs and provides insights into modern large model scaling behavior.
>
> **Q3: Noise Intensity Robustness**
> > Does different noise intensities have a big impact?
>
> Our experiments on noise types, including different non-equal noise variances, show DOT is robust to different noise intensities (Figure 5).

---

### Official Review · Reviewer_Lj6d · 2025-10-31

**Soundness:** 1
**Presentation:** 2
**Contribution:** 2
**Rating:** 2
**Confidence:** 4

**Summary:**

The paper presents a new identifiability result concerning causal DAGs and transformers: under standard assumptions, they show (1) the DAG implied by the generating structural causal model (in the common sense of Pearl, Spirtes et al., and Peters et al.) is equivalent to the Granger-causal DAG and is identifiable from the conditional independence relations and (2) the conditional independence relations are identifiable from the gradients in a standard transformer.
I find (2) and it's application in (1) quite interesting, but I'm afraid (1) is false, or at least imprecisely stated---but I think the authors can either fix it or help clear up my misunderstanding during the rebuttal.

**Strengths:**

- ground-breaking connection (if the theory holds up) between causal models and transformers
- clearly written
- good breadth of experimental settings (other than DAG sparsity) and baselines from the literature

**Weaknesses:**

- conflates the usual notion of causality with Granger-causality, especially in the title and abstract, but also elsewhere in the paper
- missing important causality context, background, and related work
- illegible figures
- limited experiments: only 3 replicates, very sparse generating DAGs, and potentially misleading metrics

**Questions:**

I'm happy to increase my score depending on how these questions are answered.
1. The paper uses just "causal" in the title and abstract and then later seems to qualify that it's only Granger-causal, and then alternates between the two notions in Section 3.1---which is it? If it really is only Granger-causal, this needs to be made much more clear, especially in the title/abstract, to avoid exaggerated claims; or if Theorem 1 really is about (nonGranger) causal DAG identifiability, it should be made more clear, because this is an important result, even without the connection to gradients. In any case, there should be more explicit discussion on the difference between the two notions.
2. How does Theorem 1 related to existing results in the literature? There should be more discussion about this, including references. As far as I know, the closest related work is [1], which suggests that Theorem 1 actually requires the stronger assumption of *conditional exogeneity* rather than the weaker causal sufficiency in A1---have I misunderstood something?
3. A4 should be discussed more carefully; since the work in [2], faithfulness isn't considered to be as mild as the discussion around L196 suggests.
4. L207: this should be "Granger-causes", shouldn't it?
5. L019: causal accuracy *does* improve with data volume for traditional methods (e.g., provably consistent algorithms, like GES and PC); are the authors rather just trying to say here that traditional methods don't scale?
6. on L247, how does complexity control in score-based methods (which is commonly, e.g., the L1 norm of the adjacency matrix) limit scalability? and maybe the word "regularization" is missing after sparsity on the sentence before?
7. L121: what does "horizons" mean here?
8. L279: precision or F1 score?
9. Fig 2 (B): "F1 averaged across"...?
10. L452: it's only "coherent" for a linear additive noise model, right? so not for transformers?
11. L492 explicitly recommends interventional data, but the method isn't at all developed for interventional data, is it?
12. L869: I wouldn't call fixed in-degree "comparable sparsity": for fixed in-degree, as the number of nodes increases, the probability of an edge existing goes to 0. For this reason, along with the unsymmetric F1 score, it's essential to include some sparse random guessing baseline in the experiments, in the experiments, to help contextualize the reported results.

[1] White, H., & Lu, X. (2010). Granger causality and dynamic structural systems. Journal of Financial Econometrics, 8(2), 193-243.

[2] Uhler, C., Raskutti, G., Bühlmann, P., & Yu, B. (2013). Geometry of the faithfulness assumption in causal inference. The Annals of Statistics, 436-463.

---

> ### Author Response · Authors · 2025-12-04
> **Response to Reviewer Lj6d**
>
> We appreciate the reviewer's insightful comments on the theoretical foundations. We address each concern below:
>
> **W1 & Q1 & Q4: Structural Causality vs. Granger Causality**
> > W1: The paper uses "causal" in the title/abstract but then qualifies it as Granger-causal—which is it?
>
> > Q4: L207: this should be "Granger-causes", shouldn't it?
>
> We use "causal" because under assumptions (A1-A4), we identify *structural causality*, edges defined by appearance in structural equations, not merely Granger causality. We have added a discussion clarifying this distinction in Appendix A.9 (Discussions, "Structural causality and Granger Causality" paragraph).
>
> Our framework is grounded in structural causal models (SCMs), where X_j → Y means X_j appears in the structural equation for Y. Under conditional exogeneity, White & Lu (2010) showed Granger non-causality equals structural non-causality almost surely. We strengthen this by: (i) imposing faithfulness for exact characterization of parents; (ii) providing a concrete computable criterion via score gradient energy; (iii) connecting this to transformers. The logical direction is from structural causality to predictive dependence.
>
> **Q2: Relationship to White & Lu (2010)**
> > How does Theorem 1 relate to existing results? White & Lu suggests Theorem 1 requires conditional exogeneity rather than causal sufficiency—have I misunderstood?
>
> **Causal sufficiency is stronger than conditional exogeneity, not weaker.**
>
> Conditional exogeneity (our A1: $U_t ⊥ X_{S^c} | X_S$) permits latent confounders that don't induce spurious dependencies; causal sufficiency (no latent confounders) is strictly stronger and trivially implies conditional exogeneity under our independent noise assumption.
>
> White & Lu show Granger non-causality equals structural non-causality *almost surely*. Theorem 1 extends this by: (1) adding faithfulness for *exact* identification; (2) providing a computable criterion via score gradient energy; (3) connecting to transformers for scalability. We clarified this hierarchy in Appendix A.1 and added discussion of relationships to prior theory in Appendix A.9.
>
> **Q3: Faithfulness Assumption**
> > A4 should be discussed more carefully; since Uhler et al. (2013), faithfulness isn't considered as mild.
>
> We assume **plain faithfulness**, not strong faithfulness in [1]. Plain faithfulness only requires that conditional independencies match d-separation and violations occur only on measure-zero hypersurfaces, and any parameter perturbation restores faithfulness, especially in nonlinear settings. We have elaborated on this in Section 3.1 and Appendix A.1.
>
> **Q5: Traditional Methods and Scaling**
> > L019: causal accuracy does improve with data volume for traditional methods; are you saying traditional methods don't scale?
>
> Yes, here we refer to behavior under heterogeneous data where traditional methods do not learn dynamic graphs and scale effectively. We have revised this sentence for clarity.
>
> **Q6: Complexity Control in Score-Based Methods**
> > On L247, how does complexity control limit scalability?
>
> The structure learned from continuous-optimization methods is highly sensitive to sparsity constraint strength. Suitable regularization is difficult to tune when graph density is unknown and there is insufficient signal. This becomes more challenging with large-scale heterogeneous data, where improper hyperparameters hurt optimization and model fitting.
>
> **Q7-10: Minor Clarifications**
> > L121 "horizons", L279 "precision or F1", Fig 2(B), L452 "coherent"
>
> Thank you for pointing these out. We have revised these passages for clarity. We also revised the details in weakness about figure settings and captions to improve readability.
>
> **Q11: Interventional Data**
> > L492 recommends interventional data, but the method isn't developed for it.
>
> That is correct, our approach does not rely on interventional data. We mention it as a validation procedure for real-world deployment.
>
> **Q12: Random Guessing Baseline**
> > Include sparse random guessing baseline to contextualize results.
>
> We would like to point out under current graph size, using expectation of incoming degree gives reasonable graph density and similarly, ER-2, ER-4 graphs are commonly used in causal discovery studies [2][3]. We have included a random guess baseline that samples binary lagged causal graphs from a Bernoulli distribution parameterized with the true graph density. We additionally provide precision, recall, and SHD in Appendix A.10.7.
>
> [1] Uhler, Caroline, et al. "Geometry of the faithfulness assumption in causal inference." The Annals of Statistics (2013): 436-463.
>
> [2] Zhu, Shengyu, Ignavier Ng, and Zhitang Chen. "Causal discovery with reinforcement learning." arXiv preprint arXiv:1906.04477 (2019).
>
> [3] Waxman, Daniel, Kurt Butler, and Petar M. Djurić. "Dagma-DCE: Interpretable, non-parametric differentiable causal discovery." IEEE Open Journal of Signal Processing 5 (2024): 393-401.

---

### Official Review · Reviewer_h6Bw · 2025-11-01

**Soundness:** 2
**Presentation:** 1
**Contribution:** 1
**Rating:** 2
**Confidence:** 4

**Summary:**

This paper claims that transformers are inherently causal learners. The argument proceeds in two parts: first, by pointing out that under causal sufficiency, faithfulness, and no instantaneous effects, causal graphs are identifiable via standard autoregression. It's then argued that transformers are well-suited, with layer-wise relevance propagation, to exploiting this. Numerical experiments are undertaken to compare to various time series structure learning algorithms, finding competitive or favorable performance in transformers according to the F1 metric. One additional contribution is experiments and arguments as to why gradient aggregation is better for graph extraction than attention weights.

**Strengths:**

- It's certainly of interest to the causality community to obtain scalable and efficient structure learning algorithms, and equally interesting to the more general ML community to understand causal properties of commonly-used architectures.
- The results are impressive, with transformers outperforming several existing causal discovery methods in various settings.
- The paper is well-written and well-motivated, which is particularly positive given the broad audience targeted.

**Weaknesses:**

The primary weakness in my opinion is the framing of the paper. To me, the paper essentially makes two separate arguments: (1) that under assumptions A1-A4, causal structure learning reduces to standard statistical learning; and (2), that transformers are particularly well-aligned to causal learning. Both of these arguments appear sound, but I would consider (1) to be well-known, and (2) to fall short of the claim that transformers are "inherently causal." Indeed, the authors concede that Theorem 1 applies to "any model that could perfectly fit the data, which means it can be any neural network architecture." While I appreciate the arguments set forth that transformers are well-aligned to (1), I believe the claim that (decoder-only, autoregressive) transformers are "inherently causal" is strong and subject to misinterpretation.

Some of the benchmarking can also be improved. For example, one could make some similar arguments that 1d-CNNs (as used in, e.g., NTS-NOTEARS) are well-suited to causal discovery (though there is, of course, evidence that transformers handle long-scale dependencies better). Including 1d-CNNs that do not enforce acyclicity constraints on instantaneous causes, or author autoregressive architectures, would help improve breadth. Including other benchmarks (for example, similar to CausalTime [1]) would improve the paper. Alternative metrics may also be considered, such as SHD.

A related weakness is the plausibility of assumptions. In particular, the lack of instantaneous edges. Indeed, in NTS-NOTEARS as an example, a significant amount of work is spent on acyclicity constraints for instantaneous causes, and much work and difficult optimization could be avoided via (A2). A potential remedy is also proposed for (A2), but unless I missed it, explicit experiments are not present.

There is also some prior work regarding the use of population gradients for graph extraction in differentiable causal discovery. NOTEARS+ [2] theoretically argues for using gradients to extract graphs, and [3] makes similar arguments to the present paper about potential pitfalls of using gradient proxies.

[1] Cheng, Y., Wang, Z., Xiao, T., Zhong, Q., Suo, J., & He, K. CausalTime: Realistically Generated Time-series for Benchmarking of Causal Discovery. In The Twelfth International Conference on Learning Representations.

[2] Zheng, X., Dan, C., Aragam, B., Ravikumar, P., & Xing, E. (2020). Learning sparse nonparametric DAGs. In International Conference on Artificial Intelligence and Statistics (pp. 3414-3425). PMLR.

[3] Waxman, D., Butler, K., & Djurić, P. M. (2024). Dagma-DCE: Interpretable, non-parametric differentiable causal discovery. IEEE Open Journal of Signal Processing, 5, 393-401.

**Questions:**

1. How do alternative architectures perform under the predictive causality framework identified via (A1-A4)?

2. How do transformers perform on more realistically-generated time series (e.g., from CausalTime)?

3. What are precision and recall rates for each method?

---

> ### Author Response · Authors · 2025-12-04
> **Response to Reviewer h6Bw**
>
> We thank the reviewer for the detailed evaluation and engaging with our theoretical claims. We respectfully address the concerns below:
>
> **W1: Novelty and "Inherently Causal" Claim**
> > Both of these arguments appear sound, but I would consider (1) to be well-known, and (2) to fall short of the claim that transformers are "inherently causal."
>
> We respectfully disagree on both points:
>
> **On (1) being "well-known":** Traditional Granger causality tests predictive improvement but does not provide unique structural identifiability, which can miss true causes or detect spurious ones. Our Theorem 1 is grounded in structural causal models (SCMs) and provides an *exact* characterization: H_j = 0 ⟺ j ∉ S. This score-gradient-energy criterion applies to a wide range of distributions (not just linear/Gaussian) and yields unique identification under standard assumptions. To our knowledge, this connection between SCM-based identifiability and gradient-based attribution is new.
>
> **On (2) and "inherently causal":** We do not claim transformers are the only capable architecture. "Inherently" means causal learning naturally happens from standard autoregressive training without explicit causal objectives or modifications. We emphasize transformers for their empirical scalability and suitability for multi-modal integration. We acknowledge that "inherently causal" may invite interpretations—our intent is to highlight that causal structure is a by-product of autoregressive transformers, not an exclusive one.
>
> **W2, Q1, Q2 & Q3: Benchmarking, Alternative Architectures, CausalTime, and Metrics**
> > W2: Including 1d-CNNs, other benchmarks (CausalTime), and alternative metrics (SHD) would improve the paper.
>
> > Q1: How do alternative architectures perform under the predictive causality framework?
>
> > Q2: How do transformers perform on CausalTime?
>
> > Q3: What are precision and recall rates?
>
> Our theoretical result (Theorem 1) is architecture-agnostic—it applies to any model that sufficiently fits the data distribution. We emphasize transformers for their scaling potential with large-scale data, multi-modal integration, and foundation model alignment in the long-term.
>
> **Alternative architectures:** We have added experiments using a 1D CNN (Appendix A.10.5). The CNN performs comparably to the deep transformer in most settings, confirming the framework generalizes beyond transformers. When data is insufficient or has low signal-to-noise ratio, smaller models with stronger inductive biases exhibit higher data efficiency.
>
> **CausalTime experiments:** We have added real-world experiments on CausalTime (Appendix A.10.3). Transformers perform comparably to state-of-the-art methods, ranking among the top performers on air quality and traffic datasets.
>
> **Additional metrics:** Precision, recall, and SHD for all synthetic experiments are provided in Appendix A.10.7.
>
> **W3: Instantaneous Edges**
> > A related weakness is the plausibility of assumptions. In particular, the lack of instantaneous edges.
>
> Although the vanilla autoregressive transformer does not model instantaneous edges, we have designed a simple and effective remedy by incorporating traditional causal discovery methods (PCMCI+ for contemporaneous relationships, L-PCMCI for latent confounders) as post-processing. See Figure 6—these procedures effectively handle assumption violations and yield more accurate recovered graphs.
>
> **W4: Prior Work on Population Gradients**
> > There is prior work regarding the use of population gradients for graph extraction (NOTEARS+, DAGMA-DCE).
>
> We thank the reviewer for highlighting these works. The key distinctions are:
>
> (1) **No DAG constraints required**: NOTEARS+ and DAGMA-DCE jointly optimize model fitting with explicit acyclicity constraints. Our approach extracts gradients from a forecasting model post-hoc. The acyclicity is naturally guaranteed by time-ordering without constrained optimization.
>
> (2) **Identification guarantees**: Our Theorem 1 establishes that gradient energy exactly identifies structural parents under faithfulness. NOTEARS+ shows gradient norms indicate dependence but relies on functional form assumptions for identifiability; DAGMA-DCE focuses on interpretability without new identification results.

---

### Official Review · Reviewer_iJgh · 2025-11-11

**Soundness:** 3
**Presentation:** 4
**Contribution:** 3
**Rating:** 8
**Confidence:** 3

**Summary:**

This paper argues that standard autoregressive transformers, when trained for time-series forecasting, implicitly learn the underlying time‐lagged causal structure. The authors build a theoretical argument—under common identifiability assumptions—that the true causal parents can be identified by examining how the model’s predictions respond to perturbations in past inputs. They propose a practical extraction method based on Layer-wise Relevance Propagation (LRP) and show that it performs competitively, often outperforming state-of-the-art causal discovery approaches, particularly in high-dimensional, nonlinear, and non-stationary settings.

**Strengths:**

- The main claim—that a regular forecasting transformer effectively learns a causal graph without any explicit causal objective—is both elegant and conceptually appealing. It links large-scale predictive modeling with causal structure learning in a way that feels natural and potentially impactful.

- The paper doesn’t rely solely on empirical results. Theorem 1 provides a clear connection between the forecasting task and recoverability of causal parents, assuming standard conditions.

- The evaluation is thorough and well thought out. The proposed method (DOT) consistently outperforms strong baselines such as PCMCI, DYNOTEARS, and VAR-LiNGAM, especially in settings that are typically most challenging: many variables, long-range dependencies, and changing distributions.

- The authors carefully examine how to extract causal structure from transformers and convincingly show why raw attention weights are not reliable, while gradient-based attribution is more informative.

**Weaknesses:**

- The method seems to require more data compared to some specialized approaches in low-dimensional or simpler linear systems. For example, VAR-LiNGAM requires far less data when the dynamics are mostly linear (Figure 3).

- (minor)  Figure 2’s caption labels two subplots as (C), which needs correction.

Missing Citations / Context


The paper compares well against classical baselines, but recent transformer-based causal discovery approaches are not discussed. For example, CausalFormer (Liu et al., 2024) also uses transformer models with attribution-based interpretation. It would be helpful to clarify how this work differs, especially given that the authors emphasize not modifying the transformer architecture.

Additionally, other recent methods aimed at high-dimensional scalability (e.g., approaches based on dynamical community partitioning or low-rank structure, constrain satisfaction with answer set programming) could provide more context for the scalability claims. Since scalability is a key selling point here, a brief discussion would help readers see the contribution in a broader landscape.

**Questions:**

You mention that the current framework (due to assumption A2) does not handle instantaneous effects but could be combined with other algorithms. Could you elaborate on how you see this working? Would you first use the transformer to get the lagged graph and then apply a method like LiNGAM on the transformer's prediction residuals, as hinted at in the paper?

---

> ### Author Response · Authors · 2025-12-04
> **Response to Reviewer iJgh**
>
> We sincerely appreciate the reviewer's positive assessment and helpful suggestions. We respond to each point below:
>
> **W1.**
> > The method seems to require more data compared to some specialized approaches in low-dimensional or simpler linear systems.
>
> We acknowledge that in small-data scenarios, statistical methods with domain priors and proper algorithm selection and hyperparameter tuning may sometimes be more efficient. We view this as a tradeoff for a more expressive and scalable algorithm, analogous to the comparison between ResNet and ViT.
>
> **W2.**
> > (minor) Figure 2's caption labels two subplots as (C), which needs correction.
>
> Thank you for pointing this out. We have corrected it.
>
> **W3 (Missing Context).**
> > The paper compares well against classical baselines, but recent transformer-based causal discovery approaches are not discussed. Since scalability is a key selling point here, a brief discussion would help readers see the contribution in a broader landscape.
>
> We thank the reviewer for highlighting the missing context on transformer-based causal discovery approaches. Our key differences can be summarized as follows:
>
> (1) **No architectural modifications**: Unlike CausalFormer, which adds multi-kernel causal convolution and decomposition-based detectors, or symmetry-aware transformers that enforce temporal asymmetry, our approach uses a vanilla decoder-only transformer without any causal-specific modifications.
>
> (2) **Unsupervised learning**: Unlike supervised approaches that require interventional data or pre-defined causal labels, and meta-learning approaches like BCNP that train on synthetic data, our method learns causal structure purely from observational forecasting without any causal supervision.
>
> (3) **Theoretical identification guarantees**: We provide formal identifiability results (Theorem 1) connecting gradient-based attributions to structural causal parents under standard assumptions, which is not established in prior transformer-based methods.
>
> We have added a discussion of these related works in Appendix A.9 (Discussion, paragraph "Transformer-based causal discovery"). We have also added a discussion about scalability in causal discovery in Appendix A.9 (Discussion, paragraph "Scalability in causal discovery") to provide a broader landscape.
>
> **Q1.**
> > You mention that the current framework (due to assumption A2) does not handle instantaneous effects but could be combined with other algorithms.
>
> We thank the reviewer for raising this limitation of the vanilla transformer architecture. We observe that instantaneous relationships in autoregressive learning act similarly to latent confounders, introducing spurious edges into the learned causal structure. Based on this observation, we propose a simpler and more systematic approach to handle instantaneous relationships by using a traditional causal discovery method that can handle such cases (PCMCI+) as a post-processing procedure, with the combination of the recovered structure and fully connected contemporaneous links as the initial skeleton. This effectively improves causal discovery quality under certain conditions (see Figure 6 and related analysis of it).

---

### Official Review · Reviewer_xm8b · 2025-11-12

**Soundness:** 3
**Presentation:** 3
**Contribution:** 2
**Rating:** 6
**Confidence:** 3

**Summary:**

This paper proposes that decoder-only transformers trained autoregressively can inherently recover lagged causal structure in time-series data. The authors theoretically show that under standard assumptions (causal sufficiency, no instantaneous effects, adequate lag coverage, and faithfulness), the gradient sensitivities of a transformer’s predictions with respect to past inputs correspond to causal dependencies. They operationalize this approach with Layer-wise Relevance Propagation (LRP) and demonstrate empirically that it outperforms classical and neural causal discovery methods on synthetic datasets spanning nonlinear, nonstationary, and high-dimensional regimes.

**Strengths:**

1. The paper is well motivated, connecting two areas - causal discovery and transformer-based autoregressive modeling. The authors clearly argue that transformers, trained with standard forecasting objectives, may implicitly learn lagged causal structures.

2. The experiments cover a wide range of synthetic regimes, including high-dimensional, long-range, nonlinear, and non-stationary systems.

3. The transformer-based approach scales efficiently to higher dimensions and larger lag windows, where traditional methods tend to time out or degrade in accuracy. The model shows monotonic improvement with sample size, indicating favorable scaling behavior.

**Weaknesses:**

1. While the paper establishes a theoretical link between population gradients and causal identifiability, it does not provide empirical evidence that gradient-or LRP-based attributions reliably correspond to true causal effects. All experiments evaluate graph recovery accuracy but do not include diagnostic analyses verifying whether gradient magnitudes align with known interventional or conditional-independence causal measures. As a result, it remains unclear whether the model’s attributions capture genuine causal mechanisms or merely reflect predictive correlations that the transformer has learned. Demonstrating empirical alignment between gradient signals and interventional effects (e.g., via controlled ablation or counterfactual perturbation tests) would be essential to substantiate the causal claims.

2. The assumptions (no confounders, no instantaneous effects) are strong and only briefly discussed in terms of practical violations.

3. The reliance on LRP is heuristic; attribution stability under noise, scaling, or initialization is not explored.

4. Sensitivity to hyperparameter top-k binarization is missing.

5. The identifiability result relies on causal sufficiency and the absence of instantaneous effects, conditions rarely satisfied in realistic systems. While these are discussed qualitatively, there is no empirical study of robustness when assumptions are violated (e.g., latent confounders, contemporaneous interactions).

**Questions:**

1. How sensitive are the recovered graphs to model depth, attention heads, or normalization layers?

2. Can pretrained forecasting transformers be reused for causal extraction without retraining?

3. Could you discuss whether gradient-based attributions correlate with intervention-based effects?

4. The proposed Top-k and uniform-threshold binarization rules control graph density differently. How is k selected in practice, and how sensitive are the F1 results to this choice?

5. Since all experiments are synthetic, to what extent do you expect the observed scaling behavior and robustness to generalize to real-world nonstationary data, where causal mechanisms may change over time?

---

> ### Author Response · Authors · 2025-12-04
> **Response to Reviewer xm8b**
>
> We thank the reviewer for the careful review and constructive feedback. We address each point below:
>
> **W1 & Q3: Gradient-Based Attributions and Intervention Effects**
> > W1: All experiments evaluate graph recovery accuracy but do not include diagnostic analyses verifying whether gradient magnitudes align with known interventional or conditional-independence causal measures.
>
> > Q3: Could you discuss whether gradient-based attributions correlate with intervention-based effects?
>
> All experiments are conducted on synthetic data with full control over data generation and ground truth causal graphs that represent genuine causal mechanisms. We have added an intervention effect analysis in Appendix A.5. We find that gradient-based attributions generally correlate with intervention-based effects: the more accurate the structure the transformer learns, the higher the correlation. In linear cases with nearly perfect structure learning, the correlation coefficient is almost perfect; in nonlinear cases with less accurate structure, the coefficient is smaller. The model attributes higher relevance scores to edges with genuine causal mechanisms.
>
> **W2 & W5: Assumption Violations and Robustness**
> > W2: The assumptions (no confounders, no instantaneous effects) are strong and only briefly discussed in terms of practical violations.
>
> > W5: While these are discussed qualitatively, there is no empirical study of robustness when assumptions are violated.
>
> The robustness analysis for latent variables is shown in Figure 5. We have added experiments on instantaneous relationships. While the vanilla transformer does not model latent variables or instantaneous relationships, we now provide practical post-processing strategies combining transformers with traditional causal discovery algorithms (L-PCMCI for latent confounders, PCMCI+ for instantaneous effects). See Figure 6 and analysis—these procedures effectively refine the learned causal graph when assumptions are violated.
>
> **W3: LRP Attribution Stability**
> > The reliance on LRP is heuristic; attribution stability under noise, scaling, or initialization is not explored.
>
> The adoption of LRP is grounded in the theoretical connection between structural causal identification and gradient energy, not heuristic. Each experiment is conducted with three random initializations, and our experiments include variations in variable dimensions, sample sizes (scaling), noise types, and noise standard deviations.
>
> **W4 & Q4: Top-k Sensitivity Analysis**
> > W4: Sensitivity to hyperparameter top-k binarization is missing.
>
> > Q4: How is k selected in practice, and how sensitive are the F1 results to this choice?
>
> We have added sensitivity analysis in Appendix A.10.4 on three linear and nonlinear datasets. We find the F1 score remains relatively stable across a reasonable range of k values, with a tradeoff between precision and recall. In practice, domain experts may leverage knowledge about system sparsity to select k, starting with smaller k for higher precision. From our uncertainty analysis (Appendix A.10.6), the model shows noticeably higher edge strength for correctly identified edges, guiding practitioners to focus on the most reliable edges via continuous heatmaps rather than only binarized graphs.
>
> **Q1: Model Architecture Sensitivity**
> > How sensitive are the recovered graphs to model depth, attention heads, or normalization layers?
>
> Comparisons of different transformer variants are shown in Figure 8. Deeper transformers with more attention heads generally perform better in scenarios with long-range dependencies, nonlinear relationships, and non-stationary environments.
>
> **Q2: Pretrained Transformer Reuse**
> > Can pretrained forecasting transformers be reused for causal extraction without retraining?
>
> We investigate zero-shot causal discovery with time-series foundation models (Chronos-2) in Appendix A.10.1. Current foundation models do not achieve perfect forecasting on unseen datasets, resulting in suboptimal structure recovery. However, fine-tuning only normalization layers and newly added node embeddings significantly improves both forecasting and structure identification.
>
> **Q5: Real-World Generalization**
> > Since all experiments are synthetic, to what extent do you expect the observed scaling behavior and robustness to generalize to real-world nonstationary data?
>
> Decoder-only transformers exhibit advantages in modeling non-stationary distributions, benefiting real-world nonstationary data, which is an advantage traditional methods lack. We acknowledge that real-world nonstationarity may introduce higher data requirements, but expect the approach to scale well as time-series forecasting techniques continue to advance.

---

### Author Response · Authors · 2025-12-04
**Review Summary**

We sincerely thank all reviewers for their constructive feedback. We are encouraged by the recognition of the **elegance and conceptual appeal** of connecting forecasting transformers with causal structure learning (Reviewer iJgh), the acknowledgment that our **theoretical framework provides a clear connection between prediction and causal identifiability** (Reviewers iJgh, Lj6d), and the appreciation for **comprehensive experiments** spanning high-dimensional, nonlinear, and non-stationary regimes (Reviewers xm8b, iJgh). Reviewer iJgh's positive assessment of our formal identifiability guarantees directly addresses novelty concerns raised by others—our gradient-based criterion provides an *exact* characterization of structural parents extending beyond classical Granger causality. While some reviewers expressed concerns about assumption plausibility, Reviewers xm8b and iJgh positively noted that our approach **scales efficiently and consistently outperforms strong baselines**. All reviewers agreed on the importance of scalable causal discovery and the potential impact of bridging foundation models with causal inference.

**Summary of Revisions:**

**Theory:** Inspired by Reviewer Lj6d's reference to White & Lu (2010), we made two extensions: (1) **Assumption relaxation**—we weakened A1 from causal sufficiency to conditional exogeneity, permitting certain latent processes; (2) **Broader identifiability**—we extended from conditional-mean gradient energy $G_j$ to score gradient energy $H_j$, ensuring identifiability under a wider range of functional forms and distributions. We extended a complete proof (A.1) and clarified the distinction between structural and Granger causality (A.9).

**New Experiments:** nonstationary settings with minimal changes (A.10.2), CausalTime real-world benchmark (A.10.3), CNN as alternative architecture (A.10.5), intervention-gradient correlation analysis (A.5), zero-shot with Chronos foundation model (A.10.1), top-k sensitivity analysis (A.10.4), uncertainty analysis (A.10.6), and complete metrics (precision, recall, SHD) in A.10.7.

**Assumption Violation Handling:** Post-processing strategies combining transformers with L-PCMCI (latent confounders) and PCMCI+ (instantaneous effects), demonstrated in Figure 6.

**Additional Context:** Discussions on transformer-based causal discovery, scalability in causal discovery and relationships to previous theory results.  (A.9).

---

**To Reviewer xm8b:** Added intervention-gradient correlation analysis, top-k sensitivity analysis, and post-processing for assumption violations. Clarified that LRP is theoretically grounded.

**To Reviewer iJgh:** Corrected Figure 2 caption, added transformer-based causal discovery and scalability in causal discovery discussions, and elaborated on instantaneous effect handling via PCMCI+.

**To Reviewer h6Bw:** Added CNN experiments, CausalTime benchmarks, precision/recall/SHD metrics, and clarified distinctions from NOTEARS+ and DAGMA-DCE.

**To Reviewer Lj6d:** Extended theoretical framework inspired by the White & Lu reference, clarified structural vs. Granger causality, elaborated on faithfulness assumptions, discussed about the relationships to previous theory results, and added random guess baseline.

**To Reviewer bY8W:** Added CausalTime experiments, uncertainty visualizations, JRNGC-F baseline comparison, and demonstrated dynamic causal graph handling.

Revisions are highlighted in **blue**. We hope these address reviewers' concerns effectively.

---

### Meta-Review · Area_Chair_PBCS · 2025-12-22

**Summary:**

This paper shows that autoregressive transformers learn information that is amenable to causal learning. I will summarize the main criticisms from the reviewers, who were split in their judgment, from negative to positive. I have also read the paper in great detail, in addition to the reviews, due to the lack of discussion and the fact that the paper had originally received extreme ratings in opposite directions.

The crux of the argument made by this paper is that causal structure can be extracted from log likelihood estimation via its derivative (i.e., the score function) in transformers trained autoregressively.

The main criticisms of the reviewers were:

xm8b: whether the score actually captures causal information vs correlations learned by the model. I think this is not a real concern in light of the theory presented by the paper and the relation to the literature I discuss later.

iJgh: Their concerns were mostly around amortized causal discovery, for which the authors have extended the discussion.

h6Bw: Their concern was that the fact that the assumptions from theorem 1 lead to identification was known, and the novelty is more around transformers being well-suited to this problem. Yet, Theorem 1 applies to any architecture, not necessarily transformers (which in my opinion was also known). I think the reviewer is right in their criticism, and the authors did not address their point sufficiently.

Lj6d and bY8W: Their main concern was between causality and Granger causality. I think the authors did a reasonable job explaining the difference, but it remains open what is specific about transformers vs autoregressive training and the relation to other methods in both Granger causality and standard structure learning.

To address the feedback of these last three reviewers in future submissions, *I would recommend* the authors to rephrase their paper, focusing on autoregressive next token prediction instead of transformers, which I think is the most novel and interesting aspect of the paper. Theorem 1 is already known (see next), but the connection with next-token prediction is novel. This also lends itself to explain well that the paper is positioned in the standard causal discovery setting, not just Granger.

I also would like to point out that I think Theorem 1 was already known. The paper, “Inference via low-dimensional coupling”, Spantini et al., JMLR 2018 showed that the score allows identification of Markov networks (i.e., the graph skeleton). “Score matching enables causal discovery of nonlinear additive noise models”, Rolland et al., 2022 already uses score matching for causal discovery, largely to identify leaf nodes, but also gave a condition to identify the parents of leaf nodes. This is very close to the setting of this paper, where nodes at time t are leafs because the transformer is “causal” (in the sense that it does not look into the future) and the assumption A2 (no instantaneous effect). These two results were put together by Montagna et al., in the two papers: “Scalable Causal Discovery with Score Matching” and “Causal Discovery with Score Matching on Additive Models with Arbitrary Noise”, both at CLeaR 2023. The first paper establishes the same condition as Thorem 1, except with the absolute value instead of the square (easy to see that one implies the other). This still assumes Gaussianity, which the latter paper takes care of. What this submission does differently is the relation with transformer training, which obviously the prior papers did not consider, and is an interesting extension. In fact, the main limitation of score matching based algorithms is that the score needs to be fitted repeatedly with an ad-hoc estimator. This problem was solved in the paper “Diffusion Models for Causal Discovery via Topological Ordering” Sanchez et al., ICLR 2023. They proposed an ad-hoc way to avoid re-fitting the score.

What I think this submission is brilliant for, is that they found a way to avoid all this custom machinery using vanilla autoregressive training. While this is more specific (i.e., it will only work on problems with this clear autoregressive structure), it’s also more elegant than the approach by Sanchez et al.

I strongly suggest that the authors look at those works and discuss them in their paper, repositioning their story.

**Reviewer Concerns:**

The reviewer's concerns are discussed in the main review. I think the authors marginally improved on some of their concerns. The paper has a positioning issue, as detailed in my meta review, which h6Bw in part called out (not clear what is specific about transformers). In fact, what is new in this paper is the formulation with autoregressive training, which is not clarified by the authors sufficiently.

**Reviewer Scores:**

See main meta review. I think the reviewers may have marginally increased their scores, but the core of the positioning of the paper was not addressed (also in part because it was not as obvious in the original reviews). In a normal year, I would have initiated this discussion *before* the end of the discussion period to give the chance to reviewers to engage with the authors on this.

---

### Decision · Program_Chairs · 2026-01-26

Reject